



# Structural, petrophysical and geological constraints in potential field inversion using the Tomofast-x v1.0 open-source code

Jérémie Giraud[1,2], Vitaliy Ogarko[3,4,†], Roland Martin[5], Mark Jessell[1,2], Mark Lindsay[1,2]

[1] Centre for Exploration Targeting (School of Earth Sciences), University of Western Australia, 35 Stirling Highway, 6009 Crawley Australia.

[2] Mineral Exploration Cooperative Research Centre, School of Earth Sciences, University of Western Australia, 35 Stirling Highway, WA Crawley 6009 Australia.

[3] The International Centre for Radio Astronomy Research, University of Western Australia, 7 Fairway, WA Crawley 6009 Australia.

[4] ARC Centre of Excellence for All Sky Astrophysics in 3 Dimensions (ASTRO 3D) Australia.

[5] Laboratoire de Géosciences Environnement Toulouse GET, CNRS UMR 5563, Observatoire Midi-Pyrénées, Université Paul Sabatier, 14 avenue Edouard Belin, 31400 Toulouse France.

[†] Formerly at Centre for Exploration Targeting (School of Earth Sciences), University of Western Australia, 35 Stirling Highway, 6009 Crawley Australia.

*Correspondence to*: Jeremie Giraud (Jeremie.giraud@uwa.edu.au)





**Abstract.** The quantitative integration of geophysical measurements with data and information from other disciplines is becoming increasingly important in answering the challenges of undercover imaging and of the modelling of complex areas. We propose a review of the different techniques for the utilisation of structural, petrophysical and geological information in single physics and joint inversion as implemented in the Tomofast-x open-source inversion platform. We detail the range of constraints that can be applied to the inversion of potential field data. The inversion examples we show illustrate a selection of scenarios using a realistic synthetic dataset inspired by real-world geological measurements and petrophysical data from the Hamersley region (Western Australia). Using Tomofast-x's flexibility, we investigate inversions combining the utilisation of petrophysical, structural and/or geological constraints while illustrating the utilisation of the L-curve principle to determine regularisation weights. Our results suggest that the utilisation of geological information to derive disjoint interval bound constraints is the most effective method to recover the true model. It is followed by model smoothness and smallness conditioned by geological uncertainty, and cross-gradient minimisation.



## 1 Introduction

Geophysical data provide information about the structure and composition of the Earth otherwise not accessible by direct observation methods, and thus plays a central role in every major Earth imaging initiative. Applications of geophysical modelling range from deep Earth imaging to study the crust and the mantle to shallow investigations of the subsurface for the exploration of natural resources. Recent integration of different

geophysical methods has been recognised as a means to reduce interpretation ambiguity and uncertainty. Further developments introduce uncertainty estimates from other geoscientific disciplines such as geology and petrophysics to produce more reliable and plausible models. Various techniques integrating different geophysical techniques have been developed with the aim to produce more geologically meaningful models, as reviewed by Parsekian et al. (2015), Lelièvre and Farquharson (2016), Moorkamp et al. (2016), and Ren and Kalscheuer

(2019), (Meju and Gallardo, 2016), and several kinds of optimization for such problems exist (Bijani et al., 2017). In the natural resource exploration sector, the calls of Wegener (1923), Eckhardt, (1940) and Nettleton (1949) for the development of comprehensive, thorough multi-disciplinary and multi-physical integrated modelling have been acknowledged by the scientific community, and data integration is now an area of active research: quoting André Revil's preface of the compilation of reviews proposed by (Moorkamp et al., 2016a): "The joint inversion

of geophysical data with different sensitivities […] is also a new frontier". The integration of multiple physical fields (both geophysical and geological) is particularly relevant for techniques relying on potential field gravity and magnetic data, as these constitute the most commonly acquired and widely available geophysical datatypes worldwide. The needs for integrated techniques is partly due to the interpretation ambiguity of geophysical data and resulting effects of non-uniqueness on inversion. Therefore, effective inversion of potential field data

necessitates the utilisation of constraints derived from prior information extracted from geological and petrophysical measurements or other geophysical techniques whenever available.

A number of methods for the introduction of geological and petrophysical prior information into potential-field inversion have been developed. For example, when limited geological information is available, the assumption is that spatial variation of density and magnetic susceptibility are collocated. This can be enforced through simple

structural constraints encouraging structural correlation between the two models using Gramian constraints (Zhdanov et al., 2012) or the cross-gradient technique introduced in Gallardo and Meju, (2003). When petrophysical information is available, petrophysical constraints can be applied during inversion to obtain inverted properties that match certain statistics (see techniques introduced by Paasche and Tronicke (2007), De Stefano et al. (2011), Sun and Li (2016, 2011, 2015), Lelièvre et al. (2012), Carter-McAuslan et al. (2015), Zhang and Revil

(2015), Giraud et al. (2016, 2017, 2019c), Heincke et al. (2017). Furthermore, when geological data are available, geological models can be derived and their statistics can be used to derive a candidate model for forward modelling (Guillen et al., 2008, Lindsay et al., 2013, de La Varga et al., 2019), to derive statistical petrophysical constraints for inversion (Giraud et al., 2017, 2019d, 2019c), and to restrict the range of accepted values using spatially varying disjoint bound constraints (Ogarko et al., 2020) or multinary transformation (Zhdanov and Lin 2017).

In this paper, we present a versatile inversion platform designed to integrate geological and petrophysical constraints to the inversion of gravity and magnetic data at different scales. We present Tomofast-x ('x' for 'extendable') as an open-source inversion platform capable of dealing with varying amounts and quality of input data. Tomofast-x is designed to conduct constrained single-physics and joint-physics inversion. The need for





reproducible research (Peng 2011) is facilitated by open-source codes (Gil et al., 2016), thus we introduce and

detail the different constraints implemented in Tomofast-x before providing a realistic synthetic application example using selected functionalities. We illustrate the use of Tomofast-x by performing a realistic synthetic study investigating several modelling scenarios typically encountered by practitioners, and provide information to get free access to the source-code and to run it using the synthetic data shown in this paper. We perform single physics inversion of gravity data and study the influence of prior information using several amounts and types of

constraints, and run joint inversion of gravity and magnetic data. The flexibility of Tomofast-x is exploited to test the effect of structural constraints combined with petrophysical and geological prior information that are yet to be demonstrated in the published literature. A challenging geological setting is used to examine the capability of cross-gradient constraints within the joint inversion method. The mathematical formulation of geophysical problems and solutions are detailed throughout the paper and sufficient information is provided to allow the

reproducibility of this work using Tomofast-x.

The remainder of the contributions revolves around two main aspects. We first review the theory behind the inversion algorithm and the different techniques used, with an emphasis on the mathematical formulation of the problem. We then present a synthetic example inspired from a geological model in the Hamersley province (Western Australia), where we investigate two case scenarios. In the first case, we apply structural constraints to

an area where geology contradicts the assumption of collocated and correlated density and magnetic susceptibility variations. In the second case, we investigate a novel combination of petrophysical and structural information to constrain single physics inversion. Finally, we place Tomofast-x in the general context of research in geophysical inverse modelling and conclude this article.

## 2 Inverse modelling platform Tomofast-x

### 2.1 Purpose of Tomofast-x

Tomofast-x can be used in a wide range of geoscientific scenarios as it can integrate multiple forms of prior information to constrain inversion and follow appropriate inversion strategies. Constraints can be applied through Tikhonov-style regularisation of the inverse problem (Tikhonov and Arsenin 1977, 1978). In single-physics inversion, these comprise model smallness (also called 'model damping', minimizing the norm of the model, see

Hoerl and Kennard 1970) and model smoothness (also called 'gradient damping', minimizing the norm of the spatial gradient of the model, see Li and Oldenburg 1996). For more detailed imaging, petrophysical constraints using Gaussian mixture models (Giraud et al., 2019c) as well as structural constraints (Giraud et al., 2019d, Martin et al., 2020), multiple interval bound constraints (Ogarko et al., 2020), can be used depending on the requirements of the study and the information available. In the case of single-physics inversion with structural constraints,

structural similarity between a selected reference model and the inverted models can be maximised using structural constraints based on cross-gradients (Gallardo and Meju 2003), locally weighted gradients in the same philosophy as Brown et al. (2012), Wiik et al. (2015), Yan et al. (2017), Giraud et al. (2019d). Generally speaking, in the joint inversion case, the two models inverted for are linked using the structural constraints just mentioned or petrophysical clustering constraints in the same spirit as Carter-McAuslan et al. (2015), Kamm et al. (2015), Sun

and Li (2015, 2017), Zhang and Revil (2015), Bijani et al. (2017). In addition to the underlying assumptions defining the relationship between properties jointly inverted for, prior information from previous modelling or





geological information can be incorporated in inversion using model and structural covariance matrices by assigning weights that vary spatially. In such case, Tomofast-x allows utilising prior information extensively. Furthermore, Tomofast-x allows the use of an arbitrary number of prior and starting models enabling the investigation of the subsurface in a detailed and stochastic-oriented fashion. Tomofast-x was initially developed for application to regional or crustal studies (areas covering hundreds of kilometres), and retains this capability. The current version of Tomofast-x is now more versatile as development is now directed toward use for exploration targeting and the monitoring of natural resources (kilometric scale).

Lastly, in addition to inversion, Tomofast-x offers the possibility to assess uncertainty in the recovered models. The uncertainty assessments include: statistical measures gathered from the petrophysical constraints; posterior least-squares variance matrix of the recovered model (in the Least Squares with QR-factorization algorithm – LSQR – sense of Paige and Saunders 1982, see 2.5), and the degree of structural similarity between the models (for joint inversion or structurally constrained inversion). From a practical point of view, associated with the inversion algorithm is a user manual covering most functionalities and a reduced 2D Python notebook illustrating concepts (see Sect. 7 for more information) that can be used for testing or educational purposes. A summary of the inverse modelling workflow of Tomofast-x is shown in Figure 1.

### 2.2 General design

The implementation we present extends the original inversion platform "Tomofast" (Martin et al., 2013, 2017). Tomofast-x is an extended implementation proposed and modified by Martin et al., 2018, Giraud et al. (2019d, 2019c), Martin et al. (2020), Ogarko et al. (2020). Tomofast-x follows the object-oriented Fortran 2008 standard and utilizes classes designed to account for the mathematics of the problem. This introduces enhanced modularity based on the implementation of specific modules than can be called depending on the type of inversion required. The utilisation of classes in Tomofast-x also eases the addition of new functionalities and permits to reduce software complexity while maintaining flexibility. Our implementation uses an indexed hexahedral solid body mesh, giving the possibility to adapt the problem geometry, allowing to regularize the problem in the same fashion as Wiik et al. (2015) or to perform overburden stripping. By default, the sensitivity matrix to geophysical measurements is stored in a sparse format (using the Compressed Sparse Row format) to reduce memory consumption and for fast matrix-vector multiplications.

Attention has also been given to computational aspects. The only dependency of Tomofast-x is the Message Passing Interface (MPI) libraries, which eases installation and usage. This allows optimal usage of multi-CPU systems regardless of the number of CPUs. Parallelization is made on the model cells using a domain decomposition approach in space. That is, the model is divided into nearly equal, non-overlapping contiguous parts distributed among the CPUs, hence enforcing minimum load imbalance. Consequently, the code is fully scalable as the maximum number of CPUs is not limited by the number of receivers or measured data points. For large 3D models, Tomofast-x can run on hundreds of CPUs for a typical problem with $10^5$-$10^6$ model cells and $10^3$-$10^4$ data points. Parallel efficiency tests reveal excellent scalability and speed performance provided that the portions of the model sent to the CPUs are of sufficient size. In the current implementation, the optimum number of elements per CPU is 512. Interested readers can refer to 0.



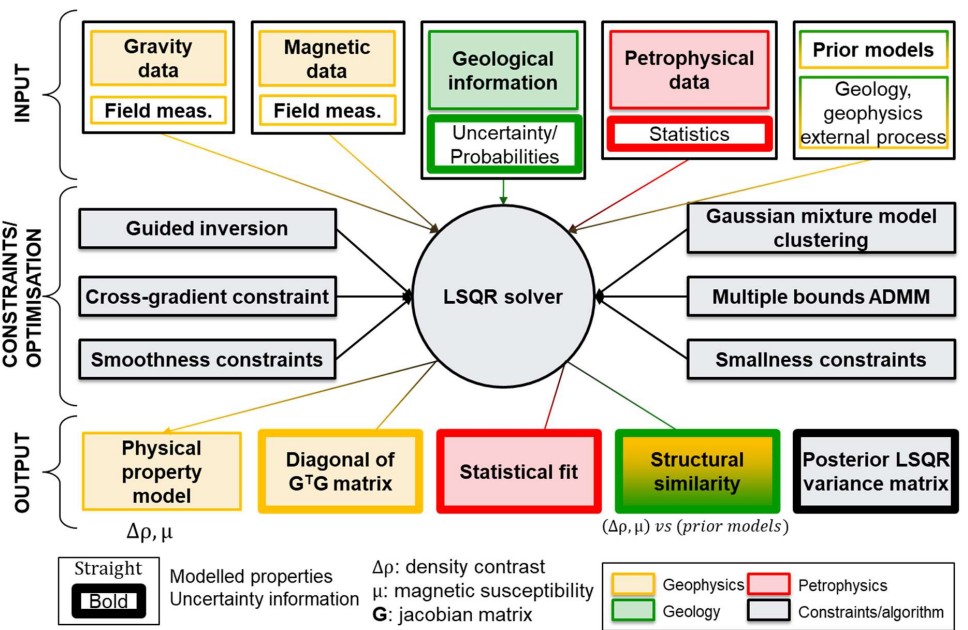

**Figure 1. Modelling workflow summary (modified from personal communication by J. Giraud).**

## 2.3 Cost function

### 2.3.1 General formulation

Tomofast-x inversions rely on optimization of a least-squares cost function and optimized iteratively. The choice of a least-squares framework was motivated by flexibility in the number of constraints and forms of prior

information used in the optimization process.

The objective function $\theta$ is derived from the log-likelihood of a probabilistic density function $\Theta$ (see Tarantola 2005, Chapters 1 and 3, for details). In the case of geophysical inversion, it is representative of the 'degree of knowledge that we have about the values of the parameters of our system' (Tarantola and Valette 1982), as summarized below. Let us first define $\Theta$ as follows:

$$\Theta(\boldsymbol{d}, \boldsymbol{m}) = \Theta_d(\boldsymbol{d}, \boldsymbol{m}) \prod_{i \in constraints} \Theta_i(\boldsymbol{m}), \tag{1}$$

where $\Theta_d(\boldsymbol{d}, \boldsymbol{m})$ is the density function over the geophysical data $\boldsymbol{d}$ that model $\boldsymbol{m}$ represents, and $\Theta_i(\boldsymbol{m})$ is the density function for the $i^{\text{th}}$ type of prior information available (the $constraints$ set).

On the premise that Gaussian probability densities approximate the problem appropriately, $\Theta_d(\boldsymbol{d}, \boldsymbol{m})$ can be expressed as:

$$\Theta_d(d, m) = A \exp\left(-\left\|\boldsymbol{W_d}(\boldsymbol{d} - \boldsymbol{g(m)})\right\|_2^2\right), A \in \mathbb{R}^+\backslash\{0\}, \tag{2}$$

where $\boldsymbol{g(m)}$ is the forward data set calculated by the forward operator $\boldsymbol{g}$; the matrix $\boldsymbol{W_d}$ weights the data points.

Similarly, we formulate the different $\Theta_{i \in constraints}$ as:



$$\Theta_i(m) = C_i \exp\left(-\alpha_i \|\boldsymbol{W}_i f(\boldsymbol{m})\|_2^2\right), C_i \in \mathbb{R}^+ \backslash \{0\}, \tag{3}$$

where f is a function of the model and prior information. $\boldsymbol{W}_i$ is a covariance matrix weighting of $f(\boldsymbol{m})$ and $\alpha_i$ contains positive scalars that are introduced to adjust the relative importance of the $i^{th}$ constraint term. $\boldsymbol{W}_i$ and $\alpha_i$ are derived from prior information, or set according to study objectives.

From equation 3, it is clear that maximizing $\Theta(\boldsymbol{d}, \boldsymbol{m})$ is equivalent to minimizing its negative logarithm, $\theta(d, m)$,
defined as:

$$\theta(d, m) = -\log \Theta(\boldsymbol{d}, \boldsymbol{m}) = \alpha_d \|\boldsymbol{W}_d(\boldsymbol{d} - \boldsymbol{g}(\boldsymbol{m}))\|_{L_2}^2 + \sum_{i \in constraints} \alpha_i \|\boldsymbol{W}_i f(\boldsymbol{m})\|_{L_2}^2, \tag{4}$$

which corresponds to the general formulation of a cost function as formalized in the least-squares framework; $\alpha_d$ is a weight controlling the importance of the corresponding data term (i.e., gravity or magnetic) in the overall cost function.

**2.3.2 Definition of regularization constraints**

Adapting the formulation of the second term of equation (4) to the different types of prior information that we can accommodate leads to the following aggregate cost function:

$$\theta(d, m) = \alpha_d \|\boldsymbol{W}_d(\boldsymbol{d} - \boldsymbol{g}(\boldsymbol{m}))\|_{L_2}^2 + \alpha_m \|\boldsymbol{W}_m(\boldsymbol{m} - \boldsymbol{m}_{pr})\|_{L_p}^2 + \alpha_g \|\boldsymbol{W}_g \nabla \boldsymbol{m}\|_{L_2}^2$$
$$+ \alpha_x \|\boldsymbol{W}_x(\nabla \boldsymbol{m}_1 \times \nabla \boldsymbol{m}_2)\|_{L_2}^2 + \alpha_{pe} \|\boldsymbol{W}_{pe} P(\boldsymbol{m})\|_{L_2}^2 + \alpha_a \|\boldsymbol{W}_a(\boldsymbol{m} - \boldsymbol{v} + \boldsymbol{u})\|_{L_2}^2, \tag{5}$$

where the different terms following the data misfit term $\alpha_d \|\boldsymbol{W}_d(\boldsymbol{d} - \boldsymbol{g}(\boldsymbol{m}))\|_{L_2}^2$ constitute constraints for the inversion of geophysical data acting as regularization in the fashion of Tikhonov regularisation (Tikhonov and Arsenin 1977). In equations (2-5), $\boldsymbol{W}_d$ represents geophysical data weighting. Generally, $\boldsymbol{W}_d$ should be the data
covariance. It is calculated by Tomofast-x as follows:

$$\boldsymbol{W}_d = \left(\sum_{i=1}^{n_{data}} (d_i)^2\right)^{-1} I_w, \tag{6}$$

where $I_w$ is a diagonal matrix equal to the identity matrix in single domain inversion, or giving the weight of one data misfit term (i.e, gravity data) against the other (i.e., magnetic data) in joint inversion; $d_i$ is the $i^{th}$ datum. By convention, we fix $I_w$ to the identity matrix $I$ for gravity inversion, and use $I_w = I\alpha_{mag}$ in joint inversion. In such cases, $\alpha_{mag}$ is a strictly positive scalar.

The main terms of the cost function are defined below. The other individual terms are defined in the next subsection and summarized in Appendix B:

- $\boldsymbol{m}_{pr}$ refers to prior model,

- $\|\boldsymbol{W}_m(\boldsymbol{m} - \boldsymbol{m}_{pr})\|_{L_p}^2$ represents the smallness term (detailed in Sect. 2.4.1);

- $\boldsymbol{L_p}$ refers to the L-p norm (here taken such that $1 < p \le 2$) ;

- $\nabla$ is the operator calculating the spatial gradient of the model;

- $\|\boldsymbol{W}_g \nabla \boldsymbol{m}\|_{L_2}^2$ represents the smoothness constraint on the model (detailed in Sect. 2.4.2);



- $\|W_x \nabla m^1 \times \nabla m^2\|_{L2}^2$ represents cross-gradient constraints between the models $m^1$ and $m^2$ (detailed in Sect. 2.4.3).
- $\|W_{pe} P(m)\|_{L_2}^2$ represents petrophysics term (clustering constraint), into which $P(m)$ represents petrophysical distributions used to impose petrophysical constraints (detailed in Sect. 2.4.4);
- $\|W_a(m - v + u)\|_{L_2}^2$ is the formulation of the multiple-bounds constraint using the alternating direction of multipliers method (ADMM, detailed in Sect. 2.4.5);

In the case of joint inversion, the vectors defined above are concatenated and the matrices expanded as follows:

$$g(m) = \begin{bmatrix} g_G(m^G) \\ g_M(m^M) \end{bmatrix}, m = \begin{bmatrix} m^1 \\ m^2 \end{bmatrix} = \begin{bmatrix} m^G \\ m^M \end{bmatrix}, d = \begin{bmatrix} d_G \\ d_M \end{bmatrix},$$

$$\alpha_{i \in [d,m,g,x,pe,a]} = \begin{bmatrix} \alpha_{i \in [d,m,g,xgpe,a]}^G & 0 \\ 0 & \alpha_{i \in [d,m,g,x,pe,a]}^M \end{bmatrix},$$

$$W_{i \in [d,m,g,x,pe,a]} = \begin{bmatrix} W_{i \in [d,m,g,x,pe,a]}^G & 0 \\ 0 & W_{i \in [d,m,g,x,pe,a]}^M \end{bmatrix}, \qquad (7)$$

where $T$ denotes the transpose operator; for more illustrative purposes of the joint inversion, we take here the gravity and magnetic joint inversion example; $G$ and $M$ refer to gravity and magnetic problems, respectively.

In the case of single domain inversion, $m^1$ is the model inverted for, and is equal to $m^G$ or $m^M$ depending on the type of geophysical data inverted, and $m^1$ is a reference model that can be used to constrain inversion from a structural point of view (see Sect. 2.4.2 and 2.4.3, and 4.4 and 4.5 for theory and example of utilisation, respectively).

In equation 5-7, subscript $m, g, x, pe$ and $a$ refer respectively to model, gradient, cross-gradient, petrophysics, and ADMM bound constraints, respectively.

As mentioned above, the cross-gradient constraints can be applied either to joint or single domain inversion. In the case of ADMM constraints, single or multiple bounds can be applied to define bounds for inverted model values. Such bound constraints can vary in space, and be made of an arbitrary number of intervals, be they disjoint or not (see Sect. 2.4.5 and 4.5). Qualitatively, the case with multiple disjoint intervals can be interpreted as applying a dynamic smallness constraint term.

In Tomofast-x, we introduce prior information in the diagonal variance matrices $W_i$ such that they are no longer homogenous and can vary in space. Note that in the implementation of gravity and magnetic inversion, $g(m) = Sm$, with $S$ the sensitivity matrix relating to measured geophysical data and corresponding recovered physical property (see Appendix C for details about their calculation). Introducing the sensitivity matrix $S_G$ and $S_M$ for gravity and magnetic data, respectively, we obtain:

$$g(m) = Sm = \begin{bmatrix} S_G & 0 \\ 0 & S_M \end{bmatrix} \begin{bmatrix} m^G \\ m^M \end{bmatrix} = \begin{bmatrix} g_G(m^G) \\ g_M(m^M) \end{bmatrix} \qquad (8)$$





For reference, the terms defined or used here are summarized in Appendix B. Tomofast-x uses the Least Squares

with QR-factorization (LSQR) algorithm (Paige and Saunders, 1982) to solve the least-squares problem. The full

matrix formulation of the problem and the related system of equations are provided in Appendix D.

Generally, not all of the terms in equation 5 are used during a single inversion. The activation of selected terms

from the cost function (setting $\alpha_i > 0$ and non-null $\boldsymbol{W_i}$) depends on the information available or on the

requirements of the modelling to be performed. For example, a term not used during inversion has the

corresponding weighting simply set to 0 (the corresponding matrix $\boldsymbol{W_i}$ is set to 0). Conversely, setting a specific

weight to a relatively large value leads to the corresponding constraint to dominate the other terms. Such practice

is typically used in sensitivity analysis to examine the effect of incorrectly assigned extreme weighting values on

the inversion by providing an example to aid detection of this unintended situation.

In the following subsection, we detail the implementation of the different terms. The terms are introduced and

detailed following the order they appear in equation 5.

### 2.4 Detailed formulation of constraints for inversion

In this Sect., we introduce the mathematical formulation of constraints applied during inversion. Throughout this

paper, *geological information* relates to information extracted from probabilistic geological structural modelling.

Petrophysical information relates to the statistics of the values inverted for (density contrast and magnetic

susceptibility).

### 2.4.1 Smallness term

We repeat the smallness term:

$$\left\| \boldsymbol{W_m}(\boldsymbol{m} - \boldsymbol{m_{pr}}) \right\|_{L_p}^2. \tag{9}$$

The smallness term corresponds to the ridge regression constraint, or smallness term of (Hoerl and Kennard,

1970). To simplify the problem, the covariance matrix $\boldsymbol{W_m}$ is assumed to be a diagonal matrix. In Tomofast-x, it

is used to adjust the strength of the constraint either globally (i.e, $\boldsymbol{W_m} = \boldsymbol{I}$) or locally (i.e., the elements of $\boldsymbol{W_m}$ may

vary from one cell to another). In the second case, $\boldsymbol{W_m}$ can be determined using prior information such as

uncertainty from geological modelling, or models and structural or statistical information derived from other

geophysical techniques (e.g., seismic attributes, probabilistic results from magnetotellurics).

### 2.4.2 Smoothness term

The smoothness model term (Li and Oldenburg 1996), is a total variation (TV)-like regularisation term based on

an original idea of Rudin et al. (1992). It constrains the degree of structural complexity allowed in the inverted

model. We repeat the term:

$$\left\| \boldsymbol{W_g} \nabla \boldsymbol{m} \right\|_{L_2}^2. \tag{10}$$

The covariance matrix $\boldsymbol{W_g}$ modulates the importance of the term by assigning the weights to each cell. For the

sake of simplicity, the matrix $\boldsymbol{W_g}$ is commonly assumed to be a diagonal matrix. It is commonly set as the identity



matrix ($\boldsymbol{W}_g = I$), but several works vary the values in space accordingly with prior information. For instance, Brown et al. (2012), Yan et al. (2017) use seismic models to calculate such weights for the inversion of electromagnetic data, and Giraud et al. (2019a), who present an application case using Tomofast-x, invert gravity

data using geological uncertainty information to calculate $\boldsymbol{W}_g$. In Tomofast-x, it can be set either globally (i.e, $\boldsymbol{W}_g = I$) or locally (i.e., the elements of $\boldsymbol{W}_g$ may vary from one cell to another).

### 2.4.3 Cross-gradient

The cross-gradient constraints were introduced as a means to link two models that are inverted jointly by encouraging structural correlation between them (Gallardo and Meju, 2004). We refer the reader to Meju and

Gallardo (2016) for a review of applications using this technique. We repeat the term below:

$$\left\| \boldsymbol{W}_{xg} (\nabla \boldsymbol{m^1} \times \nabla \boldsymbol{m^2}) \right\|_{\mathbf{L_2}}^2. \tag{11}$$

The matrix $\boldsymbol{W}_{xg}$ modulates the importance of the term by assigning the weights to each cell. In previous works, it is always (to the best our knowledge) set as the identity matrix ($\boldsymbol{W}_{xg} = I$), to the exception of Rashidifard et al. (2020), who define such weights using seismic reflectivity and apply this approach to single physics inversion of gravity data constrained by fixed seismic velocity. In Tomofast-x, three finite difference numerical schemes can

be chosen to calculate the cross-gradient derivatives: forward, centered, and mixed. In what follows, we use a 'mixed' finite difference scheme, where inversion iteration with an odd number use a forward scheme and even numbers backward scheme (e.g., iteration 3 will use a forward scheme and iteration 4 a backward scheme). This scheme was chosen as it reduces the influence of the border effects of both the forward and backward schemes onto the inverted model.

### 2.4.4 Statistical petrophysical constraints

One strategy to enforce the petrophysical constraints using statistics from petrophysical measurements is performed by encouraging the statistics of the recovered model to match that of a statistical model derived either from measurements made from the study area or literature values. In the current implementation of Tomofast-x, a mixture model representing the expected statistics of the modelled rock units is used. We use a Gaussian mixture

model to approximate the petrophysical properties of the lithologies in the studied area. In the mixture model, the weight of each Gaussian can be set in the input. We suggest to use the probability of the corresponding rock unit when this information is available. The mismatch between the statistics of the recovered model and the mixture model is minimized in the optimization framework following the same procedure described in Giraud et al., 2018, 2019b.

In the $i^{th}$ model-cell, the likelihood term $\mathrm{P}(m_i)$ of model-cell $m_i$ is calculated as, for the $k^{th}$ lithology:

$$\mathcal{N}_k = \omega_k \mathcal{N}(m_i | \boldsymbol{\mu_k}, \boldsymbol{\sigma_k}) \tag{12}$$

$$\mathrm{P}(m_i) = -\log \left( \sum_{k=1}^{n_f} \mathcal{N}_k \right) + \log \left( \max \mathcal{N}_{k=1..n_f} \right), \tag{13}$$



where:

$$\begin{cases} \omega_k = \dfrac{1}{n_f} \, everywhere \; in \; the \; absence \; of \; spatially - varying \; prior \; information \; (a) \\ \qquad\quad \omega_k = \psi_{k,i} \; \; in \; the \; i^{th} \; cell \; using \; prior \; information \; (b) \end{cases} \tag{14}$$

and $\mathcal{N}$ symbolises the normal distribution and $n_f$ is the total number of rock formations observed in the modelled area.

In practice, an expectation maximisation algorithm (McLachlan and Peel, 2000) can be used to estimate the mean $\boldsymbol{\mu_k}$ and standard deviation $\boldsymbol{\sigma_k}$ of the petrophysical measurements.

In equation 12-14, $\omega_k$ is the weight assigned to the Gaussian distribution representative of the petrophysics of the $\boldsymbol{k^{th}}$ lithology in the mixture. In equation 14a, the weight $\omega_k$ assigned to each lithology is constant across the model, while in equation 14b, the weight $\psi_{k,i}$ is derived from information derived from another modelling
technique (geology, seismic, electromagnetic methods, etc.) and varies spatially.

We note that a small number of Gaussian distributions might not be suitable to approximate certain types of distributions like bimodal (magnetic susceptibility) or lognormal (electrical resistivity) distributions. However, we point out that increasing the quality of such approximation depends on the number of Gaussian distributions used for approximation (McLachlan and Peel, 2000).

**2.4.5 Dynamic bound constraints using the ADMM algorithm**

The objective of the dynamic bound constraints is to optimize eq. 5 while ensuring that, in every model cell $m_{1 \le i \le n_m}$, the inverted value lies within the prescribed bounds such that $m_i \in B_i$, defined as ((Ogarko et al., 2020)):

$$B_i = \bigcup_{l=1}^{L_i} [a_{i,l}, b_{i,l}], \text{with } b_{i,l} > a_{i,l}, \forall \, l \in [1, L_i], \tag{15}$$

where $a_{i,l}$ and $b_{i,l}$ are the lower and upper bounds for the $i$th model-cell, and $l$ is the lithology index; $L_i \le n_f$ is
the total number of bounds allowed for the considered cell, corresponding to the number of distinct rock units allowed by such constraints. During inversion, such multiple bound constraints on physical property values inverted for are gradually enforced using the ADMM algorithm. Implementation details are beyond the scope of this paper, but more information is provided in Appendix D and we refer the reader to Ogarko et al. (2020) for details. Details about the general mathematical formulation of the ADMM algorithm can be found in Dykstra
(1983), chapter 7 in Bertsekas (2016) and Boyd (2011).

Note that the application of the ADMM bound constraints can be interpreted as are analogous to clustering constraints where (taking the example of the $k^{th}$ model-cell):

1. the centre values depend on both the current model $\boldsymbol{m}$ at any given integration and petrophysical information defining $a$ and $b$;
2. the weight assigned to each centre values changes from one iteration to the next as a function of the distance between $m_k$ and the closest bound and the number of iterations $m_k$ has remained outside $B_k$.





### 2.4.6 Depth weighting and data weighting

To balance the decreasing sensitivity of potential field data with the depth of the considered model cell, Tomofast-x offers the possibility for the calculation of the depth weighting operator. The first one, which we use in this paper utilizes the integrated sensitivities technique following Portniaguine and Zhdanov (2002). For each model cell $i$, a weight $D_{ii}$ is introduced:

$$D_{ii} = \left( \sum_{k=1}^{n_{data}} (S_{ki})^2 \right)^{\frac{1}{2}} \qquad (16)$$

The second option relies on the application of an inverse depth-weighting power law function following Li and Oldenburg (1998) and Li and Chouteau (1998) for gravity, and Li and Oldenburg (1996) for magnetic data:

$$D_{ii} = \left( \frac{1}{z_i + \varepsilon} \right)^{\beta} \qquad (17)$$

Where $z_i$ is the depth of the $i^{th}$ model cell and $\varepsilon$ is introduced to ensure numerical stability, such that $\mathbf{z} \gg \boldsymbol{\varepsilon}$; the value of $\beta$ depends on the type of data considered (gravity or magnetic). For more details about the use of depth weighting and selection of values of $\beta$, the reader is referred to the references provided in this subsection. The application of depth weighting as a preconditioner to the matrix system of equation solved for during inversion is shown in Appendix D.

### 2.5 Posterior uncertainty metrics

Uncertainty information is an important building block of modelling and a critical aspect of decision making (Scheidt et al., 2018). When available, uncertainty information can be communicated and used in subsequent modelling or for decision making (see examples of Giraud et al.2020b, 2020a, Ogarko et al., 2020, who use uncertainty information in the model recovered by another method as input to their modelling using Tomofast-x) . Tomofast-x allows the calculation of metrics reflecting the degree of uncertainty in the models before and after inversion. It allows monitoring the evolution of the different terms of the cost function during inversion. Tomofast-x also calculates uncertainty metrics that are specific to the kind of constraints used in inversion:

- the posterior covariance matrix of model $m$ as estimated by the LSQR algorithm (see Kostina et al., 2009, for details and A.1.1 for a brief introduction);
- the value of the cross-gradient in each cell;
- the individual $\mathcal{N}_k$ values (eq. 12) of the different Gaussians making up the Gaussian mixture used to define the petrophysical constraints.

The implementation of this series of indicators was performed with the intent to provide metrics for posterior analysis in detailed case studies. More information about these posterior uncertainty indicators is provided in Appendix A, which details functionalities of Tomofast-x not explored here.



### 3 Synthetic model and data


In this Sect., we introduce how the data used for synthetic modelling were derived and we present examples of using prior information derived from geological modelling. The process of simulating a realistic field case study is described with the design of the numerical experiment.

### 3.1 Geological framework

The original geological model is based on a region in the Hamersley province (Western Australia). It was built using the map2loop algorithm (Jessell. et al., n.d.) to parse the raw data and the Geomodeller® implicit modelling engine for geological interpolation (Calcagno et al., 2008, Guillen et al., 2008) to model the contacts, stratigraphy and orientation data measurement in the area (see geographical location in Figure 2). Data used to generate the model comprise the 2016 1:500 000 Interpreted Bedrock Geology map of Western Australia
(https://catalogue.data.wa.gov.au/dataset/1-500-000-state-interpreted-bedrock-geology-dmirs-016, last accessed on 02/12/2020) and the WAROX outcrop database (https://catalogue.nla.gov.au/Record/7429427, last accessed on 02/12/2020). Geological modelling was assisted by interpretation of the magnetic anomaly grid compilations at 80 meters and the 400 meters gravity anomaly grid from the Geological Survey of Western Australia (https://www.dmp.wa.gov.au/Geological-Survey/Regional-geophysical-survey-data-1392.aspx, last accessed on 02/12/2020). More information about data availability is provided in Sect. 7.

In what follows, we use an adapted version of the original structural geological framework of the selected region by increasing the vertical dimensions of the model cells and assuming a flat topography. The resulting reference geological model used to generate the physical properties for geophysical modelling measurements is shown in Figure 2 in terms of its geological units.

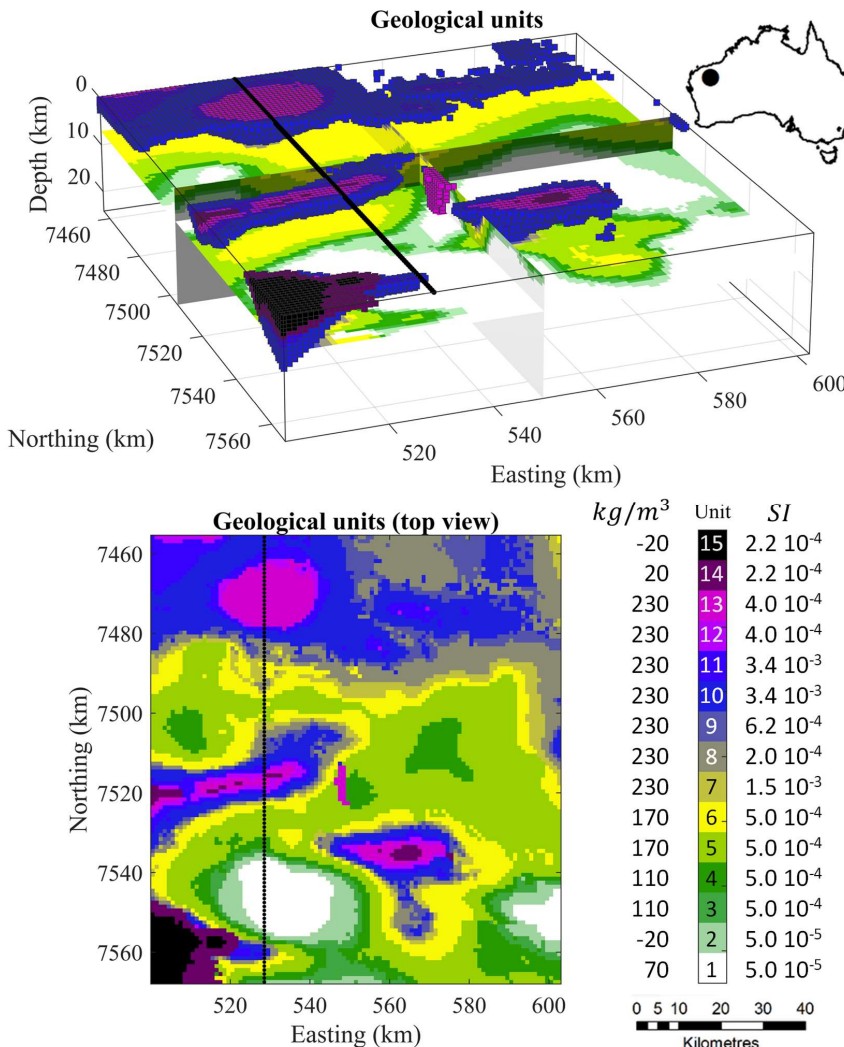

**Figure 2. True model used for geologic modelling and geophysical inversion. The top part shows the map view. The black line represents the surface coordinates of the 2D profile considered here for illustration of Tomofast-x's utilisation. The values given on either side of the colour bar indicate the density contrast and magnetic susceptibility attached to each rock unit. Note that several rock units present similar density contrast or magnetic susceptibilities, making them undistinguishable using either gravity or magnetic inversion.**

In addition to the modification of the structural model, we make adjustments on the original density values derived from field petrophysical measurements by reducing the differences between the density contrasts of different rock units. We increase the interpretation ambiguity of inversion results and decrease the differentiability of the different rock units. The same procedure is applied to magnetic susceptibility to make accurate imaging using inverse problem more challenging. The three-dimensional (3D) density contrast and magnetic susceptibility models used to generate the gravity and magnetic data are shown in Figure 3.

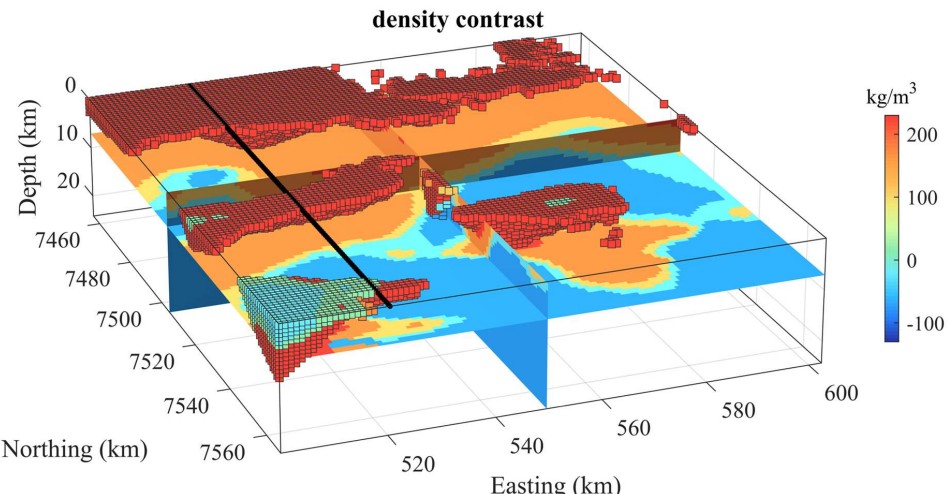

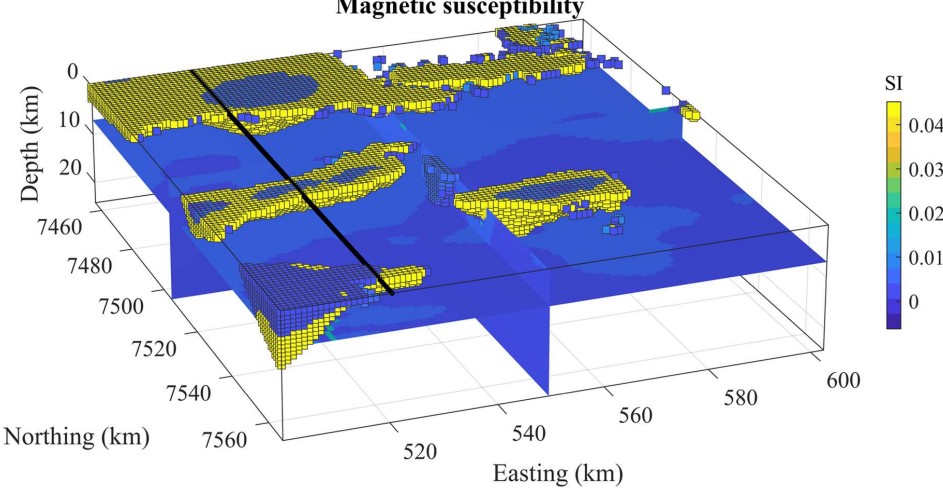

**Figure 3.** True synthetic density contrast (top) and magnetic susceptibility (bottom) model used for the simulation of geophysical data. The black line represents the surface coordinates of the 2D profile considered here. The voxels represent lithologies 10 through 15, color-coded with their respective density contrast and magnetic susceptibility values.

### 3.2 Geophysical simulations and model discretization

The core 3D model is discretised into $N_x \times N_y \times N_z = 103 \times 113 \times 33$ cells of dimensions equal to $999 \times 996 \times 745$ m³. For both gravity and magnetics, we generate one geophysical measurement per cell along the horizontal axis, leading to $N_x \times N_y = 11{,}639$ data points for each method, and add 10 padding cells in each horizontal direction to limit numerical border effects in the forward calculation, leading to a total of $123 \times 133 \times 33 = 539{,}847$ model-cells. We simulate a ground gravity survey by locating the measurements 1 m above ground





level, and aeromagnetic data acquired by a fixed-wing aircraft flying 100 meters above surface. To test the
robustness of our inversion code to noise content in the data, the geophysical data inverted here are contaminated
by noise.

The noise component was generated as follows. For each gravity measurement, we first add a perturbation value
randomly sampled from the standard normal distribution of the whole dataset multiplied by 9% of the
measurement's amplitude. We then add a second perturbation value randomly sampled from the standard normal
distribution with an amplitude of 3 mGal (2% of the dynamical range). These values were derived manually to
obtain a realistic noise contamination. To simulate small-scale spatial coherence in the noise generated in this
fashion, we then apply a two-dimensional Gaussian filter to the 2D noise map obtained from the previous step.
We then apply a two-dimensional median filter to the resulting noise-contaminated gravity data to simulate de-
noising. For magnetic data, we apply the same procedure, using, 12.5% of the measurement's amplitude for the
first step, and 15 nT (1% of the dynamical range) for the second step. Similarly to gravity data, these values were
derived manually; no levelling noise was simulated. For comparison, the noise-free and contaminated synthetic
measurements are shown in Figure 4. The resulting noise standard deviation $\sigma_{noise}$ for gravity and magnetic data
are equal to 1.2 mGal and 8.5 nT, respectively.

The gravity data modelled here correspond to the complete Bouguer anomaly. Magnetic data are simulated using
the magnetic strength of the Hamersley province (53011 nT, which approximates the International Geomagnetic
Reference Field in the area) reduced to the pole.

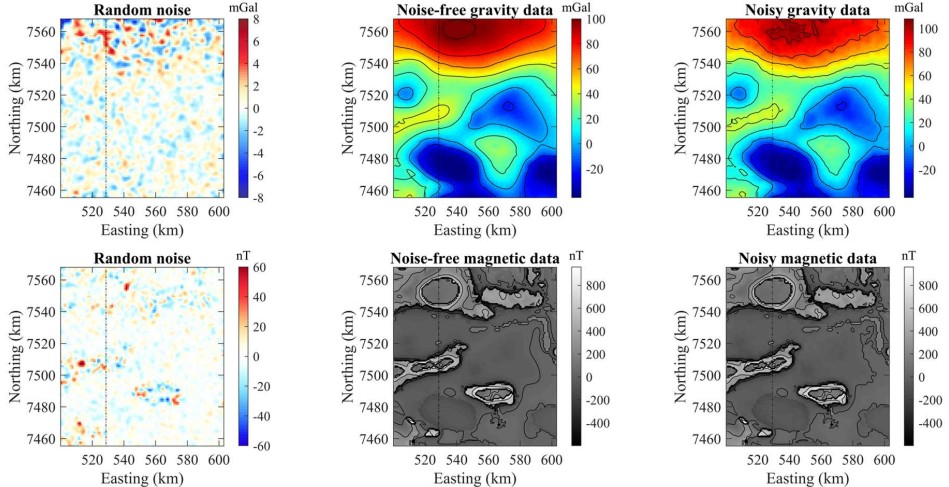

**Figure 4. Noise (left hand side column) added to the data calculated from the true model (central column) and resulting
noisy data (right hand side) for the gravity (top) and magnetic (bottom) datasets. The contour lines shown
corresponding to the ticks shown in the palette's colour bar. The black line represents the location profile we use for
the inversions performed here.**

To complete the 3D modelling procedure, a series of 100 structural geological models are generated using Monte-
Carlo perturbations of the geological measurements (foliation and contact points between geological units)
constraining the geological structures. This was performed using the Monte Carlo Uncertainty Estimator (MCUE)
technique of Pakyuz-Charrier et al. (2018, 2019). The result is an ensemble of models that all fit the geological



measurements within a given set of prior uncertainty parameters. The ensemble is thus assumed to represent the geological model space, rather than just a single 'best-guess' model. Probabilities for the occurrence of different rock units can be calculated from the ensemble and used to constrain geophysical inversion (see examples of Giraud et al., 2017, 2019d, Ogarko et al., 2020). More specifically, MCUE is useful to obtain the probability $\psi_{i,l}$

of occurrence of the different lithologies $l$ for every $i^{th}$ model cell, and to calculate the related uncertainty indicators (Sect. 3.3). Detailing the probabilistic geological modelling procedure and analysing the results in 3D is beyond the scope of this paper and interested readers are referred to Lindsay et al. (2012), Pakyuz-Charrier et al. (2018), Wellmann and Caumon (2018) and references therein.

In this contribution, we simulate a case study where modelling is carried out along the 2D profile materialized by
the black line in Figure 3 and Figure 4, extracted from the 3D modelling framework as detailed in the next subsection.

### 3.3  2D model simulation in a 3D world

As mentioned above, we perform the inversion of geophysical data along a 2D profile for simplicity and to simulate the challenging case of 2D data acquired in a 3D geological setting, in a part of the model where
subhorizontal or gently dipping features can be observed.  The philosophy of the numerical study presented here is summarized in Figure 5.

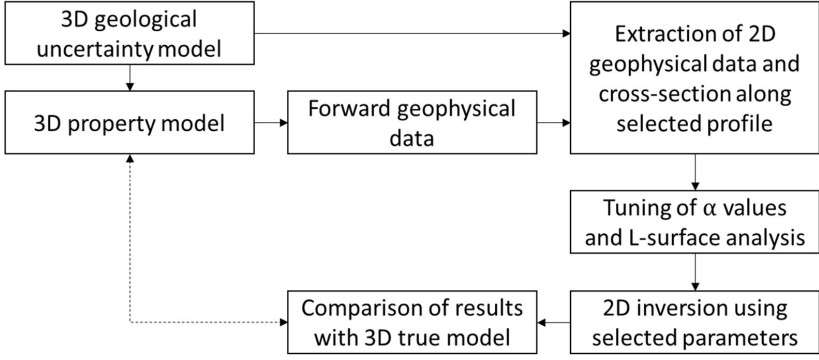

**Figure 5. Summary of experimental protocol for synthetic study and testing of different functionalities of Tomofast-x.**

The 2D geological Sect. considered is shown in Figure 6 (the black line marked in Figure 3 and Figure 4).
Geological certainty is estimated using a measure of the dispersion away from the perfectly uninformed case where the all rock units are equiprobable. In each model-cell, this measure, which we write $\boldsymbol{\sigma}'_{\psi}$, is calculated as a function of the standard deviation $\sigma_{\psi}$ of the probability $\boldsymbol{\psi}$ of observing the different rock units, as follows:

$$\boldsymbol{\sigma}'_{\psi} = \sqrt{\frac{1}{\boldsymbol{card}}} - \boldsymbol{\sigma}_{\psi}, \qquad (18)$$

where $\boldsymbol{card}$ is the geological cardinality of the model and is equal to the number of possible rock units observed in one location across the entire ensemble. From equation 18, $\boldsymbol{\sigma}'_{\psi}$ is maximum where rock units are well





constrained, and minimum where the model is the most uncertain. This geological certainty metric is shown in
Figure 6 for the 2D Sect. considered in this example.

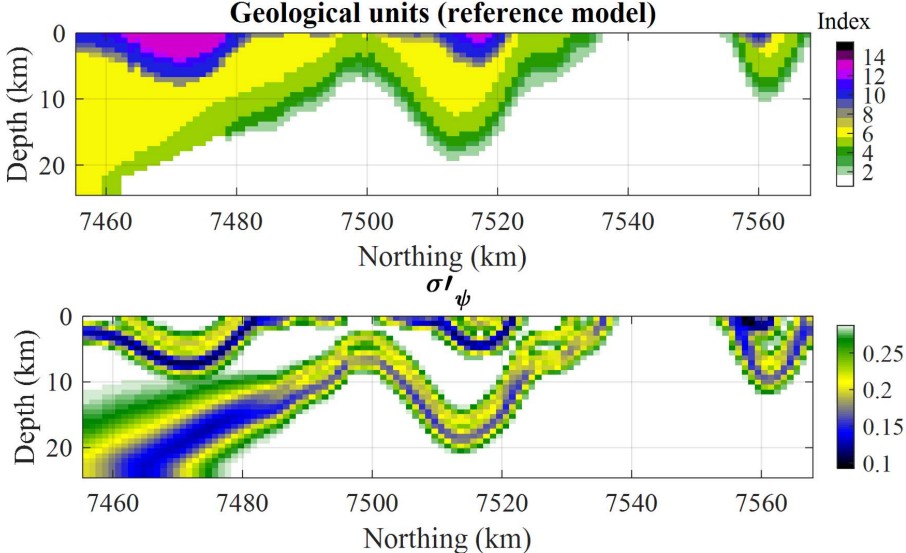

**Figure 6. Two-dimensional slice extracted from the 3D model along the profile: geological reference model (top), the**
$\sigma'_\psi$ **metric (bottom). Here, the geological uncertainty metric considered is the standard deviation of the probability of**
**the different lithologies as per equation 18.**

The probabilities of observation of the different lithologies are shown in Figure 7. Note that for the purpose of the
tests we perform on gravity data inversion, we reduce the set of probabilities by grouping rock units into fictitious
units with the same density contrasts as single rock units. This reduces the number of rock units to six units that
can be distinguished by gravity inversion as several units may be assigned the same density contrast.

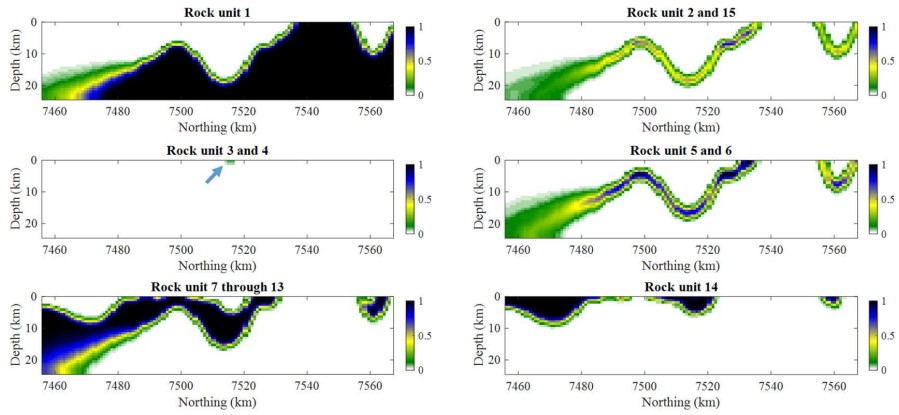


**Figure 7. Observation probabilities for the rock units that present differencing density contrasts. Units 3 and 4 are**
**nearly absent from the Sect (see maximum probability area marked by the arrow).**



The geophysical data we use for inversion are extracted along the line marked in Figure 3 and Figure 4. The geophysical data and reference (true) petrophysical model extracted in this fashion are shown in Figure 8. Care was taken not to use the same mesh for both generating the synthetic dataset and its inverse modelling.

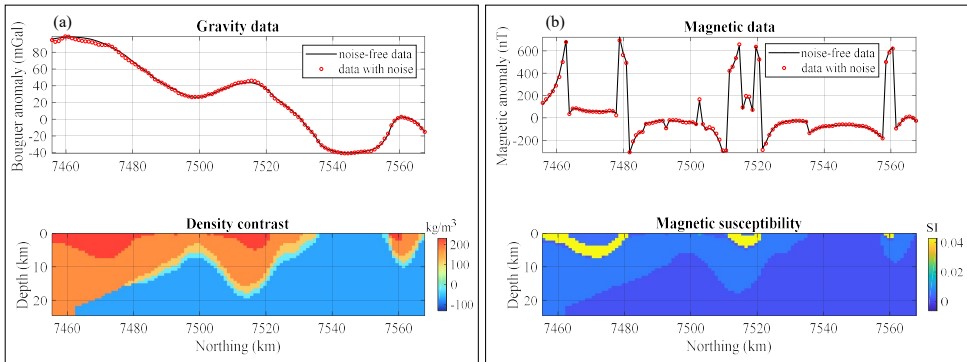

**Figure 8. Two-dimensional slice extracted from the 3D model: gravity data and density contrast (a), magnetic data and magnetic susceptibility (b).**

To inverse model the data shown in Figure 8, we generate a mesh centred on the profile (oriented along the y-direction) and add padding to either side and the northern and southern extremities. The resulting model comprises $n_x \times n_y \times n_z = 13 \times 133 \times 33$ cells of dimensions equal to $2998 \times 996 \times 745$ m. Note that we increased model-cells' size in the direction perpendicular to the profile. The gravity and magnetic datasets along the profile each comprise 113 data points evenly distributed along the line.

While we focus on a 2D section extracted from a 3D model presented here (see location in Figure 3 and Figure 4), the 3D model and the associated gravity and magnetic datasets shown here are made publicly available (see Sect. 7).

## 4 Application example: sensitivity analysis to constraints and prior information

In the examples shown below, we first perform single domain and joint (multiple domain) inversion (using the cross-gradient constraint) assuming identity matrices for $W_m$, $W_g$, and $W_x$. We then investigate the influence of prior information on single domain inversion by combining structural and petrophysical information in the case of gravity inversion. The combination of petrophysical and structural constraints derived from geology is tested. The intention is to address knowledge gaps in the literature that describes the effects of parameterization of such constraints.

### 4.1 Experimental protocol

It is necessary to determine the appropriate weights α assigned to the terms defining the constraints applied during inversion to optimize the cost function in equation 5. The α values that define the weights of the different terms in the cost function constitute hyperparameters of the inverse problem. Appropriate estimation of these hyperparameters is necessary to approximate the optimum value of the global misfit function. To this end, we use the L-curve principle (Hansen and O'Leary 1993, Hansen and Johnston 2001, Santos and Bassrei 2007) for each





of the cases presented below. We perform series of inversions sampling α values spanning the plausible range of potential choices using a heuristic approach.

When two constraint terms are used in inversion (i.e., with α > 0), we extend the L-curve approach to the two-parameter cases. In such case, the optimum values for the α weights are determined applying the L-curve criterion using L-surfaces (or elbow surface) instead of L-curves (Belge et al., 2002) (we note that the L-curves as plotted here can also referred to as 'Tikhonov curves' in the case where data misfit is plotted as a function of regularisation

value). The optimum value for the α values of the two constraint terms is therefore obtained by identification of the inflection point of the surface made up of the variations of the data misfit as a function of the weights under consideration. See Farquharson and Oldenburg (2004) for a general introduction and Giraud et al.(2019b), Martin et al. (2020), Wang et al.(2020)for application of this principle to inversions using Tomofast-x. The role the L-

surface analysis plays in the synthetic case presented here is reminded in the workflow shown in Figure 5. In our analysis, we set the objective value for the data misfit $\left\|\boldsymbol{W_d}(\boldsymbol{d} - \boldsymbol{g}(\boldsymbol{m}))\right\|_2^2$ to be equal to the objective data misfit $\Theta_d^{obj}(\boldsymbol{d}, \boldsymbol{m})$, defined as:

$$\Theta_d^{obj}(\boldsymbol{d}, \boldsymbol{m}) \geq \frac{n_{data}\sigma_{noise}}{\sum_{i=1}^{n_{data}}(d_i)^2},$$ (19)

so that the data is reproduced with a level of error superior or equal to the estimated noise level of the data. Here, this leads to $\Theta_d^{obj} = 5.01 \times 10^{-4}$ for gravity inversion, and to $\Theta_d^{obj} = 1.55 \times 10^{-4}$ for magnetic data inversion,

respectively.

For the sake of consistency in our study of the influence of constraints onto inversion, we set $\boldsymbol{m_{pr}} = 0$ kg/m³ for gravity data inversion and $\boldsymbol{m_{pr}} = 0$ SI for magnetic data inversion.

**4.2 Homogenously constrained potential field inversions**

We first perform single physics inversion following the common strategy of constraining the model using

smallness and smoothness constraints. Obtaining a good approximation of the optimum values of these parameters gives insights into the numerical structure of the problem. It constitutes valuable knowledge when using other kinds of constraints and we consider it a good practice to run such inversion prior to using more advanced constraints. Here, the first α parameters to determine are $\alpha_m$ and $\alpha_g$, for both gravity and magnetic data inversion, assuming identity $\boldsymbol{W}_m$ and $\boldsymbol{W}_g$ matrices so that the constraints are applied homogenously over the entire model.

We generate grids in the $(\alpha_m, \alpha_g)$plane using $\alpha_m \in [10^{-8}, 10^{-6}]$ and $\alpha_g \in [10^{-6}, 10^{-3}]$ for gravity inversion, and $\alpha_m \in [10^3, 10^5]$ and $\alpha_g \in [10^3, 10^8]$ for magnetic data inversion, respectively. These ranges were determined empirically and assumed to comprise the optimums. In this subsection, all matrices $\boldsymbol{W}$ in equation (5) a the identity matrix.



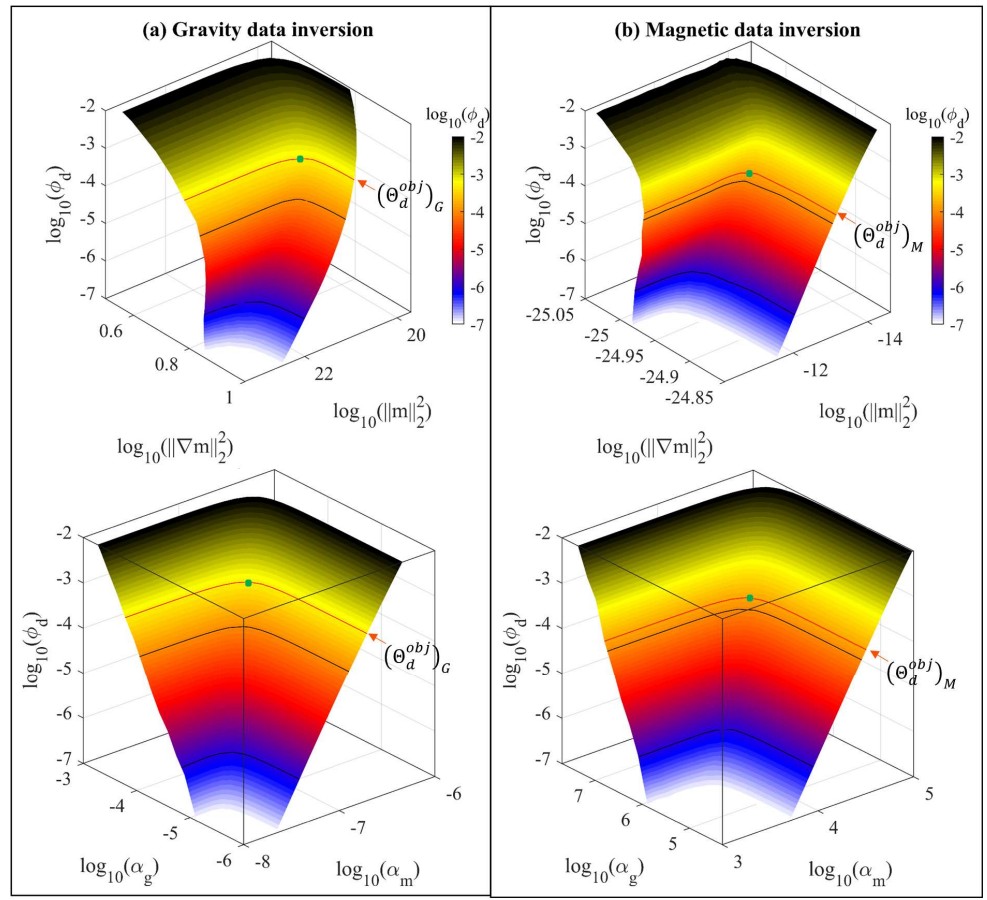

**Figure 9. Elbow surfaces for gravity (a) and magnetic (b) inversions (top) and plot of the data misfit term as a function of the $\alpha$ weights (bottom). Each plot uses a total of 1260 points sampling the $(\alpha_m, \alpha_g)$ plane. The black lines show the contour values corresponding to the ticks shown in the palette's colour bar, which shows the value of the data misfit term. The red line materialises the contour value of $\Theta_d^{obj}$, guiding the selection of the optimum $(\alpha_m, \alpha_g)$ values, and the green dot materializes the vicinity of the curve's inflection point.**

For accurate estimation, the $(\alpha_m, \alpha_g)$ values are sampled more finely closer to the estimated optimum values. The resulting L-surfaces are shown in Figure 9, where the vicinity of the optimum value of $(\alpha_m, \alpha_g)$ is shown with a green dot. From these values, we estimate the optimum values of $(\alpha_m, \alpha_g)$ reported in Table 1.

**Table 1. Optimum values of $(\alpha_m, \alpha_g)$ estimated from L-surface analysis.**

|  | $\alpha_m$ | $\alpha_g$ |
|---|---|---|
| Gravity inversion | $2.1 \times 10^{-7}$ | $1.8 \times 10^{-4}$ |
| Magnetic inversion | $3.22 \times 10^4$ | $4.52 \times 10^6$ |

The models corresponding to values in Table 1 are shown in Figure 10.

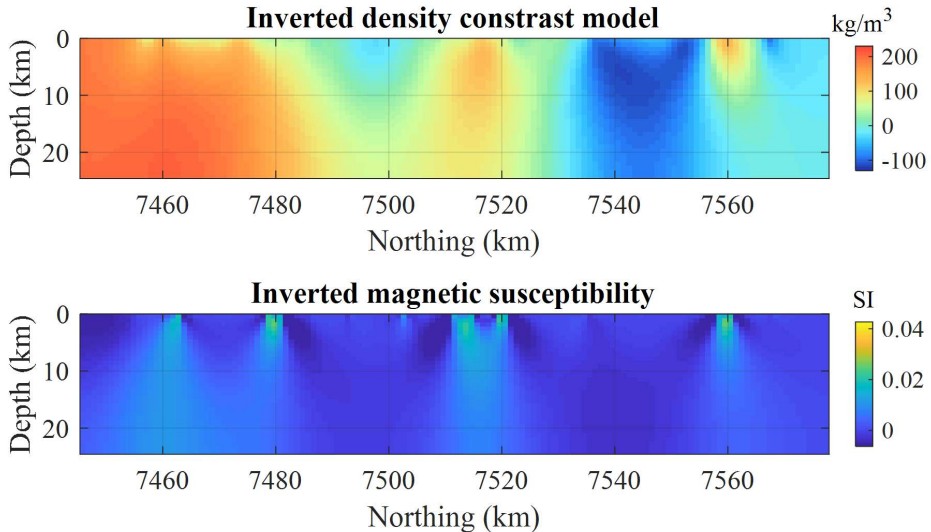

**Figure 10. Results from separate inversions using smallness and smoothness constraints. The starting and prior models model are equal to zero everywhere and the smoothness constraint is applied homogeneously.**

The values of $\alpha_m$ and $\alpha_g$ obtained for such constraints can be used as a starting point in subsequent inversions to understand the influence of prior information when varying amounts and types of prior information are available about the structure of the subsurface or its petrophysics. For instance, in what follows we will investigate the utilisation of geological information to define $W_m$ and $W_g$ (eq. 9 and 10, respectively) and see how it can combined with petrophysical data to define $B$ (eq. 15) (see following subsection where we use global and

structural and/or petrophysical information). In the case of structural constraints relying on the spatial derivatives of model values (cross-gradient or local smoothness), the value of $\alpha_m$ may be kept constant and while the other $\alpha$ parameters ($\alpha_g$ or $\alpha_x$) are adjusted. Conversely, $\alpha_g$ may be kept and $\alpha_m$ set to 0 for the utilisation of petrophysical constraints acting on the model values themselves instead of the spatial derivatives (ADMM or statistical petrophysical constraints). Here, we restrict our analysis to two $\alpha$ values being strictly superior to zero, thereby

accounting for prior information in up to two constraints terms in the definition of the regularisation term in eq. 5.

**4.3 Joint inversion using the cross-gradient constraint**

We start from the previous step to perform joint inversion using the cross-gradient constraint. Keeping the $\alpha_m$ weight constant and equal to the values determined from single domain inversion, it remains necessary to estimate

the optimum values of the cross-gradient constraint weight, $\alpha_x$, and the relative importance given to the gravity and magnetic data misfit terms (setting $\alpha^G = 1$, it remains to determine $\alpha^M$). We therefore investigate values in the ($\alpha_x, \alpha^M$) plane, which we sample in the same fashion as in the previous sub-section. The resulting surfaces are shown in Figure 11.



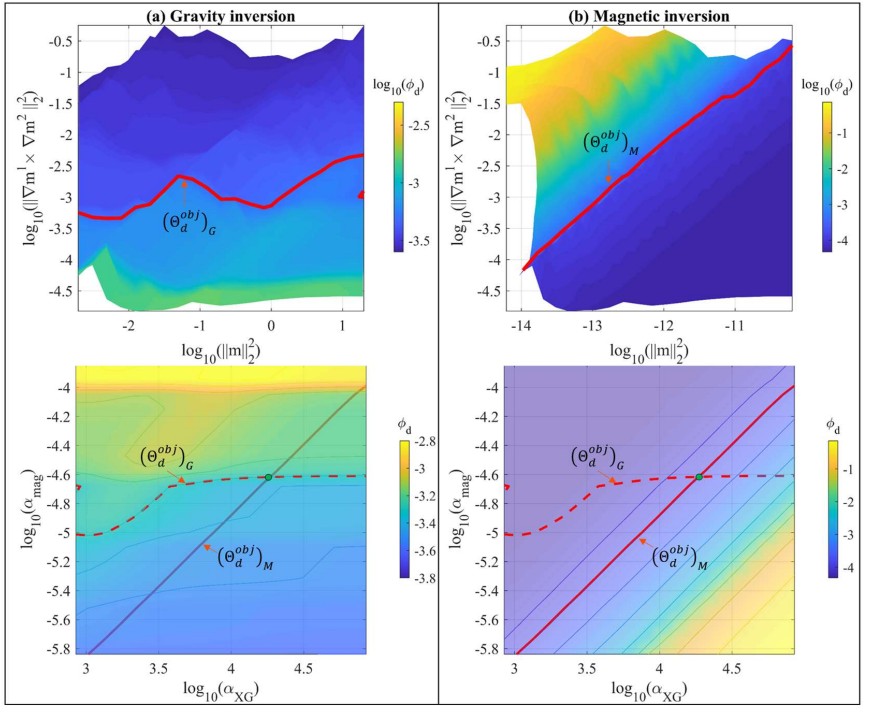

**Figure 11. Determination of the optimum $(\alpha_{xg}, \alpha^M)$ parameters in the case of the joint inversion using the cross-gradient constraint. Top view of the elbow surfaces for gravity (a) and magnetic (b) inversions (top) and plot of the data misfit term as a function the $\alpha$ weights (bottom). The solid lines show the contour values of the data misfit, which values are given by the color bar on the side. The bold red solid line shows the contour level of $\theta_d^{obj}$ for the corresponding dataset (gravity or magnetic) while the dashed line shows the same quantity for the other dataset. The green dot marks their interSect., indicating the optimum $(\alpha_{xg}, \alpha^M)$ values.**

In contrast to the single-physics inversion shown in Figure 9, it appears from Figure 11 that the two hyperparameters to be determined here, $\alpha_{xg}$ and $\alpha^M$, influence the inversion differently. While the contour levels of the magnetic data misfit show a linear trend in the $(\alpha_{xg}, \alpha^M)$ plane, it is clearly non-linear in the case of gravity data misfit. This difference might be explained by the fact that the cross-gradient is a second order regularisation (product of two spatial derivatives of model values) linking two models that are otherwise decoupled. In addition, this suggests that in cases differing from this one, the hyperparameter selection may be non-unique. Nevertheless, the value of the optimum value is unambiguous in our case and can be determined easily. From Figure 11 we obtain $(\alpha_{xg}, \alpha_{mag}) = (1.995 \times 10^4, 2.57 \times 10^{-5})$. The corresponding inversion results are shown in Figure 12.



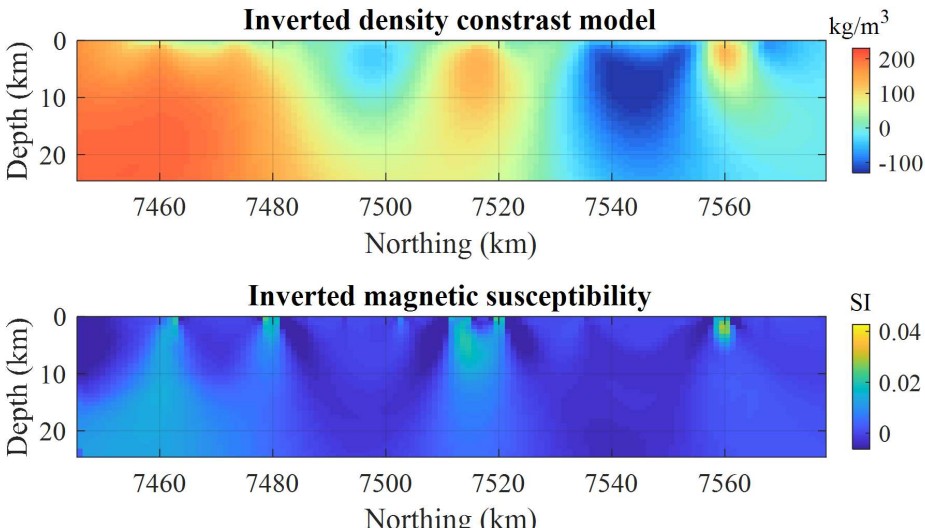

**Figure 12. Joint inversion results obtained from utilisation of the cross-gradient constraint.**

Compared to Figure 12, we observe that the application of the cross-gradient constraint leads to adjustments of the model ensuring more structural consistency between density contrast and magnetic susceptibility, illustrating the applicability of the approach presented here. Also note that the model is also visually closer to the true model from approximately 7520 km Northing and more. However, despite the increased structural consistency between the density contrast and magnetic susceptibility models, some of the structures of the model are not recovered accurately. For instance, the basin-shape structure around 7500 km Northing mirrors the actual geological structure (see Figure 8) and is an effect of non-uniqueness onto inversion. In this case, this illustrates the need for prior information in our inversion. While joint inversion of gravity and magnetic data using the cross-gradient constraint improves imaging comparatively with an inversion constrained only using smallness and smoothness constraints, prior geological information or petrophysical information may be necessary to alleviate the remaining uncertainty.

### 4.4 Smallness and smoothness constraints using geological information

In this subsection, a sensitivity analysis to prior information in inversion is performed through a series of scenarios where geological structural information is introduced to adjust the smallness and smoothness constraints through $\boldsymbol{W}_m$ and $\boldsymbol{W}_g$, respectively. In what follows, we apply this approach to gravity inversion.

The influence of geological information in defining the smallness and smoothness terms (detailed in 2.4.1 and 2.4.2) is analysed by investigating three additional scenarios allowed by the utilisation of either homogenous or geologically-derived $\boldsymbol{W}_m$ and $\boldsymbol{W}_g$ matrices. In each case, we start from the ($\alpha_m$, $\alpha_g$) weights estimated in Sect. 4.2 from the analysis of the L-surface, which we adjust to obtain the geophysical misfit sufficiently close to objective values. We remind that $\alpha_m$ and $\alpha_g$ weight the overall contribution of the model smallness and smoothness, respectively, in the cost function (eq. 5).





In the first scenario we investigate (scenario b in Table 2), geological uncertainty information is used to define $W_m$ while keeping $W_g$ homogenous. This allows us to test the influence of geological prior information onto the smallness term. The values of the diagonal variance matrix $W_m$ are calculated using the geological certainty

metric $\sigma'_\psi$ (eq. 18, shown in Figure 6b for the 2D section modelled here), and keep $W_g$ homogenous.

Contrarily to the previous tests (see 4.2) where $W_m = I_{n_m}$, we have, for the $k^{th}$ model cell:

$$(W_m)_{kk} = (\sigma'_\psi)_k. \qquad (20)$$

Because $0 \leq (\sigma'_\psi)_k \leq 1 \; \forall \, k$, we have:

$$tr(W_m) = \sum_{i=1}^{n_m} (\sigma'_\psi)_i \leq tr(I_{n_m}) = n_m. \qquad (21)$$

Consequently, setting $W_m$ in this fashion and keeping $\alpha_m$ constant translates in a lower overall relative importance of the smoothness term in the least-squares cost function (eq. 5), thereby moving away from the trade-off inferred

from the L-curve principle (Sect. 4.2). To mitigate this, we adjust $\alpha_m$ to a value $\alpha'_m$ such that:

$$\alpha'_m = \alpha_m \sqrt{\frac{n_m}{\sum_{i=1}^{n_m}(\sigma'_\psi)_i}}, \qquad (22)$$

which equates $(\alpha'_m)^2 tr(W_m)$ with $(\alpha_m)^2 n_m$ so that the overall weight assigned to the smallness term remains the same with and without geological structural information (left-hand side and right-hand side of inequality in equation 20, respectively). Because the values of $\|W_m m\|$ depend on both $W_m$ and $m$, which vary in space and also depends on the other terms of the cost function, minor adjustments of the value of $\alpha'_m$ are necessary to reach

the objective value of data misfit. In this example, this leads to tune the suggested $\alpha'_m = 8.4 \times 10^{-7}$ to $\alpha'_m = 8.85 \times 10^{-7}$ (keeping $\alpha_g$ constant). The corresponding inverted model is shown in Figure 13b. The corresponding $\alpha$ weights are repeated in Table 2.

In the second scenario we test (scenario c in Table 2), geological uncertainty information is then used to define $W_g$ while keeping $W_m$ homogenous. This allows us to test the influence of geological prior information onto the

smoothness term. Following the same procedure as for the smallness term, we adjust the suggested $\alpha'_g = 3.6 \times 10^{-4}$ to $\alpha'_g = 4.1 \times 10^{-4}$. The corresponding inverted model is shown in Figure 13c. Finally, we test the case where both $W_m$ and $W_g$ are defined using geological information in the form of $\sigma'_\psi$. Starting from values of $\alpha'_m$ and $\alpha'_g$ used in the previous tests, minor tuning is performed, leading to $\alpha'_m = 6.1 \times 10^{-7}$ and $\alpha'_g = 6.0 \times 10^{-4}$. Inversion results in the model shown in Figure 13d.

**Table 2. $\alpha$ values derived for simultaneous usage of local and global smallness and smoothness constraints. Scenario (a) is a reminder of values obtained in 4.2 when only global constraints are used.**

|  | $\alpha_m$ | $\alpha_g$ |
|---|---|---|
| Global smoothness, global smallness constraints (a) | $2.1 \times 10^{-7}$ | $1.8 \times 10^{-4}$ |
| Global smoothness, local smallness constraints (b) | $8.85 \times 10^{-7}$ | $1.8 \times 10^{-4}$ |
| Local smoothness, global smallness constraints (c) | $2.1 \times 10^{-7}$ | $4.1 \times 10^{-4}$ |
| Local smoothness, local smallness constraints (d) | $6.1 \times 10^{-7}$ | $6.0 \times 10^{-4}$ |


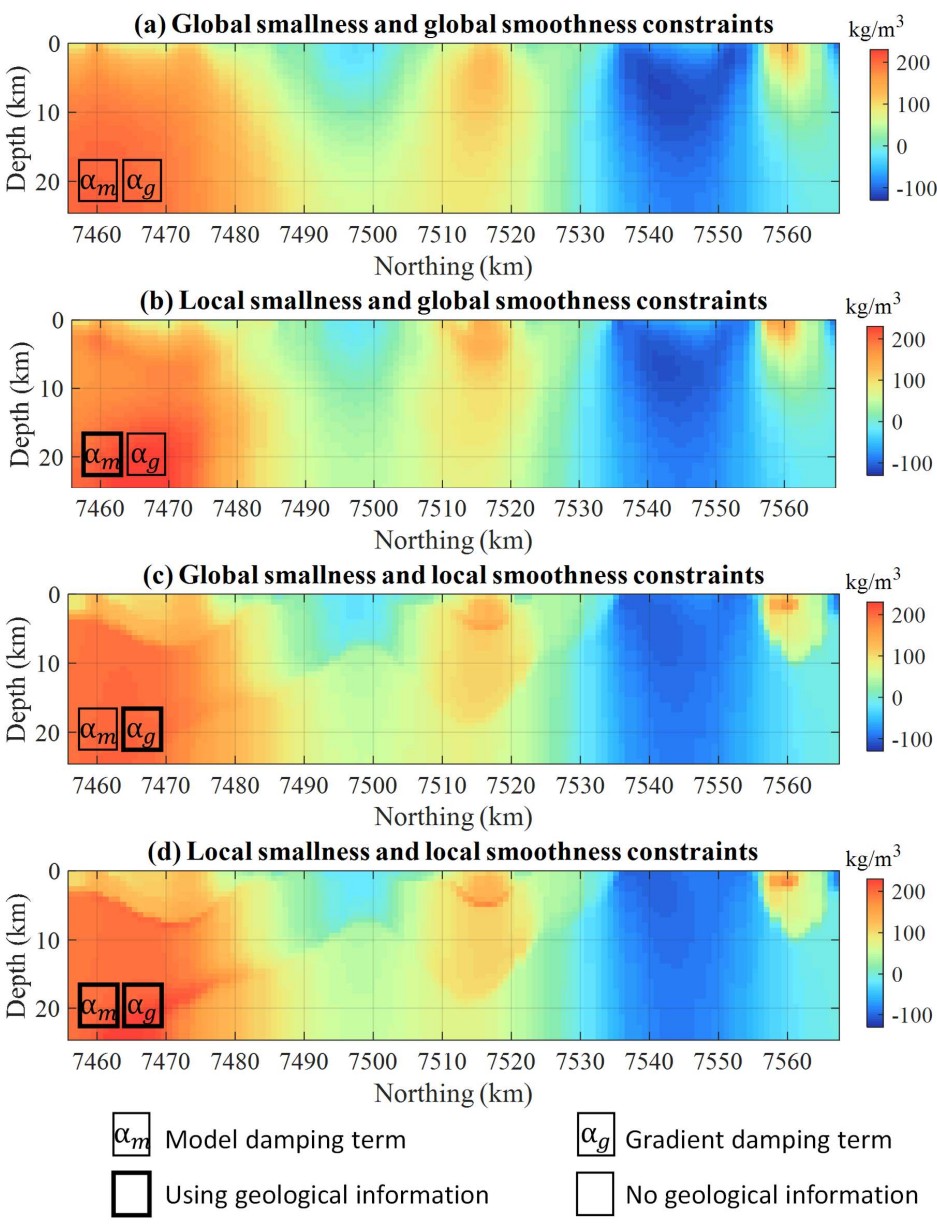

Figure 13. Results from gravity inversion constrained by: (a) homogenous smallness and homogenous smoothness constraints, (b) geologically-derived smallness and homogenous smoothness constraints (c) homogenous smallness and geologically-derived smoothness constraints, (d) geologically-derived smallness and geologically-derived smoothness constraints.



As can be seen in Figure 13 by comparing Figure 13a-b with Figure 13c-d, the utilisation of geological structural
information to adjust the smoothness regularization strength spatially has more impact on inversion than adjusting
the smallness term. While incorporating prior geological information in $\boldsymbol{W}_m$ constrains the model to a certain
extent, using $\boldsymbol{C}_v$ to derive $\boldsymbol{W}_g$ has more influence on the inverted model than for $\boldsymbol{W}_m$, with resulting models that
closer to the reference model.

Comparing Figure 13c and Figure 13d indicates that the use of geological uncertainty information to adjust the
smallness regularization strength spatially (through $\boldsymbol{W}_m$) in addition to the model smoothness term (through $\boldsymbol{W}_g$)
modifies inversion results further towards the reference model. Figure 13d, which results from inversion using
prior information in both constraint terms, provides the model closest to the reference. While most interfaces are
well-recovered when using geological information to define both $\boldsymbol{W}_g$ and $\boldsymbol{W}_m$, the recovered density contrasts
remain affected by the ambiguity inherent to gravity data in the presence of subhorizontal geological units (around
7460 km northing). This suggests that in this example, more prior information might be useful in recovering the
causative model more truthfully, especially in cases where potential field inversion is ambiguous (e.g.,
subhorizontal interfaces for gravity inversion). This is the object of the next subsection, which describes a new
single-physics inversion scenario where petrophysical constraints are combined with structural constraints and
geological information.

**4.5 Structural and petrophysical constraints**

In addition to the definition of matrices $\boldsymbol{W}_m$ and $\boldsymbol{W}_g$, geological information can be combined with petrophysical
knowledge to define the range of density values allowed in inversion. This is achieved with spatially varying
bound constraints on the property inverted for – density contrast in this case (see Sect. 2.4.5). Here, such bounds
are defined using multiple intervals, each one corresponding to the range of density contrast values expected for
a geological unit. Such bounds can be defined globally (homogenously) where all intervals are allowed
everywhere in the model, or locally when prior information about the presence of the rock units is available. In
this work, we use the probability of occurrence of the different rock units to derive bounds that vary in space
accordingly with the probability of observation of each of the rock units. In a given cell, only the bound values
corresponding to rock units with a probability $\Psi > 0$ are considered. Starting from eq. 15, such spatially-varying
bounds $B_k$ of the $k^{th}$ model-cell are obtained as follows:

$$B_k = \bigcup_{\substack{l=1 \\ \Psi_{k,l}>0}}^{n_f} [a_{k,l}, b_{k,l}], \tag{23}$$

where $a$ and $b$ correspond to lower and upper bounds. We consider narrow bounds such that $b_{k,l} = a_{k,l} + \varepsilon$, with
$\varepsilon \ll a_{k,l}$, to encourage inversion to use density contrasts that closely resemble values defined a priori. Equation
23 corresponds to the application of a Boolean operator to the probabilities $\Psi_{k=1..n_f}$ in every cell to divide the
studied area into domains defined by rock units with a probability $\Psi > \Psi_{th}$. In such case, the ADMM bound
constraints act as a proxy for a prior model dynamically constrained by petrophysical information.





Four additional scenarios are tested to determine the influence of prior information onto inversion to accommodate the addition of both the damping gradient and ADMM bound constraint term. The use of prior information is illustrated in Figure 14.

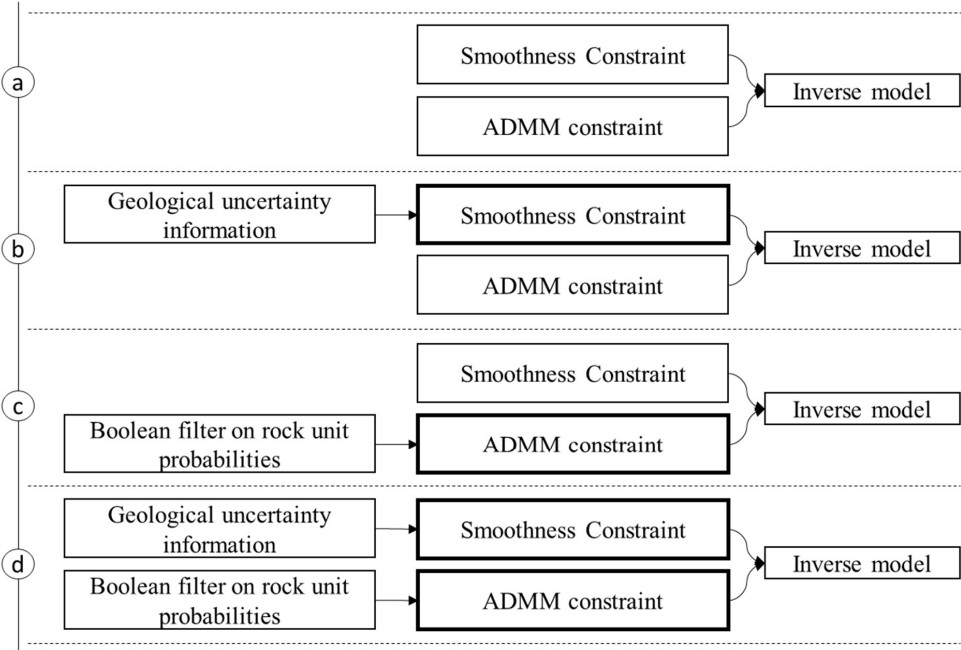

**Figure 14. Tested combinations for the utilisation of prior information into inversion. Bold frames indicate the utilisation of geological information to define the constraints.**

At a given iteration, the ADMM bounded constraint encourages inverted values to evolve inside one of the prescribed intervals depending on the current model $m$. As mentioned above, we can then make the analogy with a smallness term that is dynamically updated. For this reason, we treat the ADMM bounded constraints in the 650 same fashion as the smallness term, which we apply simultaneously to the model smoothness term.

Following the same protocol as Sect. 4.1 to determine $\alpha_g$ and $\alpha_a$, we first perform inversion without the use of geological information in the form of the probabilities for the occurrence of different rock units or metrics that can be derived from them (Figure 14a, i.e., with $\mathbf{W}_g$ and $\mathbf{W}_a$ equal to the identity matrix and the corresponding regularization term weighted by $\alpha_g$ and $\alpha_a$, respectively). It is therefore necessary to determine the value of $\alpha_g$ 655 and $\alpha_a$ (eq. 5). We perform an L-surface analysis and sample values in the $(\alpha_g, \alpha_a)$ plane to estimate the optimum values for these hyperparameters (see Figure 15). Values of $\alpha_g$ varying from $1.585 \times 10^{-7}$ and $1.585 \times 10^{-5}$, and values of $\alpha_a$ vary from $2.484 \times 10^{-5}$ and $2.484 \times 10^{-7}$. The resulting L-surface is shown in Figure 15.





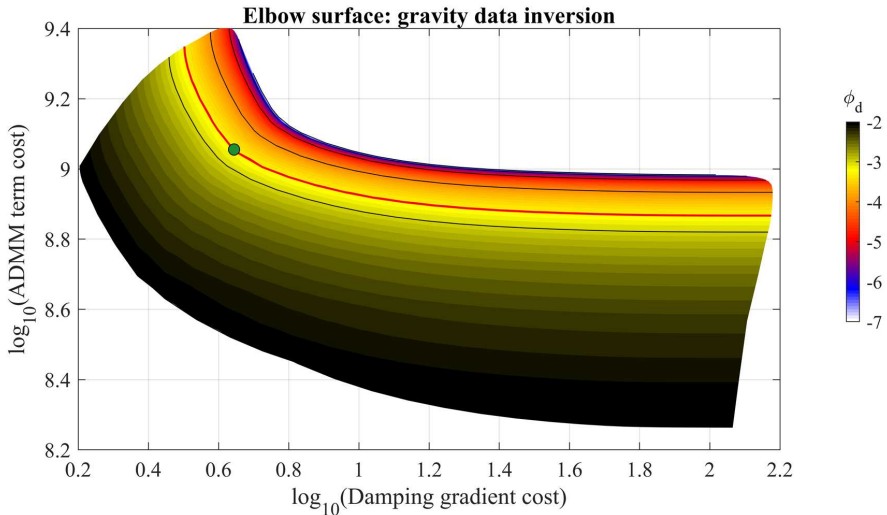

**Figure 15. Elbow surfaces for gravity inversions. A total of 840 points sampling the $(\alpha_g, \alpha_a)$ plane were used. The black**
**lines show the contour values corresponding to the ticks shown in the palette's colour bar, which shows the value of the**
**data misfit term. The red line indicates the contour value of $\Theta_d^{obj} = 5.008 \times 10^{-4}$, guiding the selection of the optimum**
**$(\alpha_g, \alpha_g)$ values, and the green point indicates the curve inflection point.**

From Figure 15, we estimate the hyper-parameters $(\alpha_g, \alpha_a)$ to be $(\alpha_g, \alpha_a) = (2.2 \times 10^{-5}, 1.3 \times 10^{-7})$ in the case
no geological information is used, meaning that both constraints are applied homogenously across the model.

From there, we follow the same procedure as described above (Sect. 4.1 and 4.2) to obtain an estimate for the
values of $\alpha_a$ and $\alpha_g$ in the different configurations shown in Figure 14b-d. The resulting inverted models are
shown in Figure 16, and the estimates of $(\alpha_g, \alpha_a)$ are provided in Table 3.

**Table 3. $\alpha$ values derived for simultaneous usage of global and local smoothness and ADMM bound constraints. Cases**
**(a) through (d) correspond to cases (a) through (d) in Figure 14.**

|  | $\alpha_g$ | $\alpha_a$ |
|---|---|---|
| Global constraints (a) | $2.2 \times 10^{-5}$ | $1.3 \times 10^{-7}$ |
| Global ADMM, local gradients (b) | $3.3 \times 10^{-4}$ | $2.6 \times 10^{-7}$ |
| local ADMM, global gradients (c) | $1.1 \times 10^{-4}$ | $3.6 \times 10^{-7}$ |
| local ADMM, local gradients (d) | $3.1 \times 10^{-4}$ | $3.25 \times 10^{-7}$ |






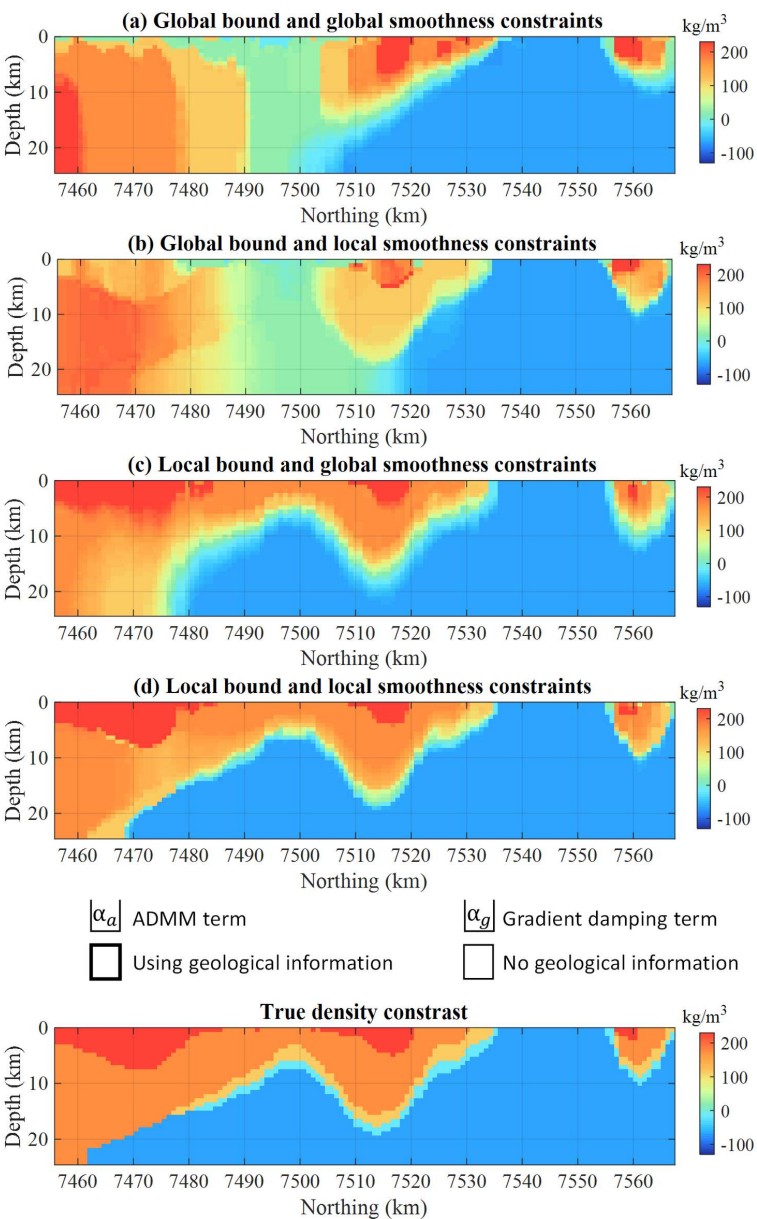

**Figure 16. Results from gravity inversion using: (a) global ADMM clustering and homogenous smoothness constraints, (b) global ADMM clustering and geologically-derived smoothness constraints, (c), ADMM clustering and geologically-derived smoothness constraints, and (d) geologically-derived ADMM clustering and geologically-derived smoothness constraints. For visual comparison, the true model is reminded at the bottom of the Figure.**


Figure 16 shows that the use of ADMM facilitates recovery of better-defined interfaces between rock units than in previous inversions (Figure 10, Figure 12, and Figure 13), and decreases the misfit with the causative model (shown in Figure 6). Unsurprisingly, without the use of geological information (Figure 16a) inversion results remain inconsistent with geology in several parts of the model, especially around the position 7500 km north. The



inconsistent results can be partly mitigated by using geologically-derived smoothness constraints (Figure 16b). In comparison, however, Figure 16c shows that use of geological information to determine the bounds recovers features much closer to the causative model.

While Figure 16d shows the more robust results overall, Figure 16c and Figure 16d present generally similar features. This indicates that, in this case, geological uncertainty information in structural constraints only allows

refining features largely controlled by the utilisation of the ADMM constraints. This statement is supported by Figure 16 where the comparison cases (a,b) and (c,d) reveal that the effect of using geological information to define bounds dominates over the effect of using uncertainty to define structural constraints.

The comparison of cases (a,b) and (c,d) in Figure 16 can be extrapolated to Figure 13 and Figure 16, to compare constraints more broadly. This is discussed in 5.1, which presents a short comparative analysis of all gravity

inversion results.

## 5 Discussion

### 5.1 Sensitivity analysis summary: comparison of constrained inversions

Tomofast-x was developed with the intent of providing practioners with an inversion platform accounting for various forms of prior information and geophysical datasets. We have tested a series of constraints involving joint

inversion, geological and petrophysical information. The inverted density contrast models for inversion using global smallness and smoothness constraints, joint inversion using the cross-gradient technique, geologically-derived smallness and smoothness constraints, and ADMM bound constraints (both global and using geological information) are shown in Figure 17.



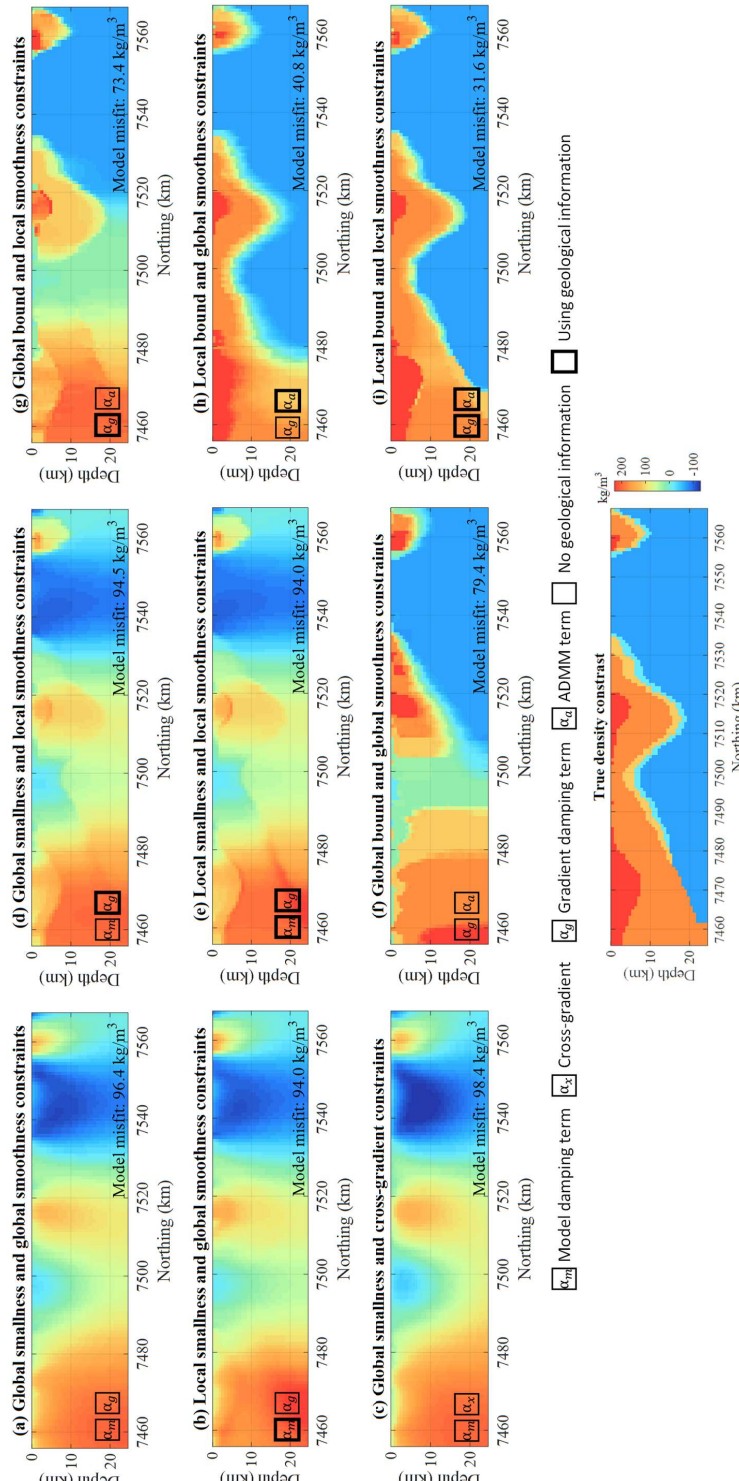

**Figure 17. General comparison of all inversion results obtained from gravity inversion. The legend on the Figure identifies the different types of inversion shown. We remind the true model at the bottom of the Figure. The model misfit indicated on each panel is calculated as the root-mean-square of the difference between the inverted and true models.**






Firstly, it appears from Figure 17 that regardless of the type of constraints considered, the utilisation of geological information (cases b, d, e, g, h, i) to derive spatially-varying constraints for the $W$ matrix of both terms used provides the models that are visually closest to the true model. In this category, the utilisation of petrophysical

information in the ADMM bound constraints provides (cases h-i) models that are closest to the true model (lowest model misfit values indicated in the titles of the panels in the Figure). Secondly, the comparison of cases (a) and (d), (b) and (e), (f) and (g), and (h) and (i) indicates that while it has a less significant influence on the results, incorporating geological information in the definition of the smoothness term also influences inversion results significantly. Lastly, comparison of cases (a) and (b), and (d) and (e) suggests that the utilisation of geological

information to adjust the smallness strength spatially has an effect on inversion that is, with the cross-gradient constraints (where structural information is passed on from another geophysical dataset), the lowest.

From the results shown in Sect. 4.2 through 4.5 and compared in Figure 17, it is possible to make a qualitative, speculative ranking of the constraints accordingly with their influence on the resulting model (from the most important influence to the least important influence):

ADMM bound constraints > smoothness constraints > smallness constraints > cross-gradient constraints. This observation is also corroborated by the values of the root-mean-square misfit between the true and inverted model.

From these observations we also deduce that when geological uncertainty information is added to the definition of constraints (i.e., $\sigma'_\psi$ for defining $W_m$ and $W_g$ and probabilities for defining $B$), the term of the cost function with the highest influence on the process will determine the main features of the model, which will be adjusted

by the other term.

Tomofast-x was developed, with the intent of providing practioners with an inversion platform allowing various forms of prior information and geophysical data. Constraints that represent uncertainty and our level of epistemic knowledge provide useful constraint to inversion. This is encouraging as the Tomofast-x platform addresses a gap in inversion schemes that rely on a single model, and that model being as similar as possible to the target region,

an often impossible requirement to meet. Thus Tomofast-x opens additional research avenues to the community that are widely acknowledged, but remain largely unaddressed. Conceptual uncertainty relating the prior assumptions made about tectonic event history of the region, and thus the structure under study can be analysed. Different event histories and topologies can be considered, giving a wider scope to the model space, and allowing the geophysics to invalidate implausible histories, but giving us pause to consider other that may be less likely,

but nonetheless possible.

## 5.2 Outlook for future developments

Another research avenue under consideration is the integration of results from probabilistic modelling of seismic and electrical data into Tomofast-x. As stated in introduction, one of the goals born in mind during the design of Tomofast-x is interoperability. Current work involves the integration of Tomofast-x into the Loop[1] open source

---

[1] https://www.loop3d.org/





3D probabilistic geological and geophysical modelling platform (Ailleres et al., 2019), in an effort to unify geological and geophysical modelling at a more fundamental level than the more common cooperative approaches. Ongoing developments include the possibility to adjust weight assigned to the bound constraints accordingly with uncertainty levels in prior information used to defined spatially varying intervals.

Future research includes the utilisation of implicit geological modelling (in the sense of Calcagno et al., 2008)
with Tomofast-x to define geological structures and rules that inversion will be encouraged to follow. It also comprises the incorporation of topological laws previously used a posteriori (Giraud et al., 2019b) directly into inversion. The electrical capacitance tomography component of Tomofast-x (Martin et al., 2018), which we have not detailed here, can be extended to acoustic/seismic or electromagnetic data inversions that rely on the resolution of similar non-linear inversion problems. It opens the door to more versatility in the code and can be applied to
joint inversion in similar ways but on more than two physical domains.

In addition, future developments comprise the collaborative and joint inversion of seismic and potential field data. It is planned to develop an interface between Tomofast-x and Unisolver (not yet open-source released by its authors), which is an extension of Seimic_Cpml codes (Komatitsch and Martin 2007, Martin et al., 2010, 2019) where integrated seismic imaging solvers are implemented. Unisolver is a multi-purpose 2D/3D seismic imaging
platform based on high order finite difference and finite volume discretization as well as nonlinear seismic data inversion procedures. Such interface would allow performing collaborative or joint inversion of seismic and gravity or magnetic data and obtain the resulting models on the same mesh while benefitting from Tomofast-x's various functionalities. This will be an easy way to provide Tomofast-x with separate seismic information like sensitivity kernels on the fly as another physical domain.

In the implementation presented here, only the truncation of the matrix system based on maximum distance thresholding was discussed. It is planned to reduce memory requirements using the wavelet compression of the matrix system of the inverse problem in the same fashion as (Martin et al., 2013).

We have shown a number of tests using a selected set of functionalities of Tomofast-x. However, more, or different, tests could be done. For instance, an interesting research avenue is to exploit Tomofast-x's capability to
read an arbitrary number of prior and starting models to test the geological archetypes that can be identified by clustering of the set of geological models probabilistic geological modelling can produce (Pakyuz-Charrier et al., 2019). Additional features of Tomofast-x which testing lies beyond the scope of this paper are Jacobian matrix truncating, the different kinds of depth weighting, and their effects on the different types of inversion. Last, we have not used posterior uncertainty indicators listed in Sect. 2.5 and A.1 as the paper focusses on the inversion
capabilities of Tomofast. The output results of Tomofast-x allow, however, to study uncertainty in the same fashion as Giraud et al. (2017, 2019c) where some of them are used.

Results obtained using the cross-gradient technique for joint inversion of gravity and magnetic data showed that it can improve imaging of geological structures. However, our study also revealed some of the limitations of this method. In the synthetic example, structurally coherent features of the resulting model contradict the geology of
the true model. In addition, our L-curve (or L-surface) analysis suggests that the determination of the optimum α weights of the cost function using the cross-gradient technique may be affected by non-uniqueness and that multiple sets of weights could equally satisfy the L-curve criterion. One interpretation is that this method remains



affected by uncertainty and could be producing several families of models fitting geophysical data equally well. This observation differs from similar analysis performed in the case of joint inversion using petrophysical constraints, where such potential ambiguity was not suggested by the L-surfaces (Giraud et al., 2019c). These impressions, however, require a more detailed investigation and constitutes a new research avenue.

In our sensitivity analysis, we have produced a series of models that can be considered geophysically equivalent because they fit the geophysical data equally well. These models are the result of deterministic inversion, where prior information guides inversion towards one of the modes of the probability density function describing the problem (equation 1), or modifies them. It is therefore safe to assume that each mode is representative of an archetype of models from the geophysical data's null-space. This highlights the interest of using 'null-space shuttles' allowing navigation of the null-space (Deal and Nolet 1996, Muñoz and Rath 2006, Vasco 2007, Fichtner and Zunino 2019) to explore the space of possible models without extensive sampling and to assess the robustness of the result. In addition, the plots of the L-curves corresponding to the problem we presented suggest the presence of multiple optima in the hyperparameter space (weights $\alpha$), which it might be interesting to investigate in future research, especially in the joint inversion case.

## 6 Conclusion

We have introduced the open source joint inversion platform Tomofast-x and demonstrated its capabilities with a realistic dataset taken from the Hamersley region in central Western Australia. The geophysical theoretical background of Tomofast-x was explained in depth to guide users in understanding and using the modelling approach implemented in the source code.

We leveraged the modularity of Tomofast-x to study the sensitivity of inversion to prior structural, geological and petrophysical information, joint inversion, and the code's scalability. We tested a new combination of constraints incorporating geological structural information in the smoothness term and dynamic prior model definition using petrophysical knowledge (ADMM bound constraints), a feature usually not available to most inversion software. Our sensitivity analyses on prior information and different constraints reveal that constraints using petrophysics (ADMM clustering bound constraints) dominate over gradient-based constraints (smoothness and cross-gradient constraints), which in turn exert more influence onto inversion than smallness constraints. This shows the importance of prior information in inversion and illustrates the need to study the space of geophysically equivalent models.

The examples described here were designed to replicate a typical, rigorous approach to the development of a geoscientific model and be relevant to real-world application. The aim to ensure rigour and reproducibility of the result presented is facilitated by the release of the source code, datasets, and a reduced, modified python version of the algorithm that accompany this paper.

## 7 Source code, documentation and data availability

The source code for Tomofast-x as used in this manuscript can be found in Ogarko et al. (2021) (https://zenodo.org/record/4454220#.YFwpEK8zZaQ). The latest versions will be available from the Github



project https://github.com/TOMOFAST/, to which access can be granted upon request. It contains two repositories: 'TOMOFASTx', containing the latest version of Tomofast-x and elements to guide users in the
utilisation of the code, and 'Tomoslow2D', a Jupyter notebook illustrating concepts used in Tomofast-x on a differing algorithm, in 2D.

The geological model, a description of the input data and the geophysical models are given in Jessell et al. (2021). It also contains a dataset using the same model projected onto a finer mesh of approximately 4,2M cells and 80,000 geophysical data. The datasets are licensed under the Attribution-ShareAlike 4.0 International (CC BY-
SA 4.0) license (see https://creativecommons.org/licenses/by-sa/4.0/legalcode for details). Tomofast-x's source code is licensed under the MIT License (https://opensource.org/licenses/MIT).

## 8 Competing interests

The authors declare that they have no conflict of interest.

## 9 Acknowledgements

Appreciation is expressed to the CALMIP supercomputing centre (Toulouse, France), for their support through Roland Martin's supercomputing projects no. P1138_2018, no p1138_2019 and for the computing time provided on the Olympe machine, and in part from the TOSCA 2018-2019 grants from the CNES (French National Space Agency) . VO acknowledges the Australian Research Council Centre of Excellence for All Sky Astrophysics in 3-D (ASTRO 3-D) for supporting some of his research efforts. JG, ML and MJ are supported, in part, by Loop –
3D Enabling Stochastic 3D Geological Modelling (LP170100985) and the Mineral Exploration Cooperative Research Centre (MinEx CRC) whose activities are funded by the Australian Government's Cooperative Research Centre Program. This is MinEx CRC Document 2021/3. ML acknowledges funding from the ARC and DECRA DE190100431.

The authors acknowledge the contribution of Clement Barriere and Ashwani Prabhakar to the reduced 2D Python
version of Tomofast-x (called 'Tomoslow') and to the writing the user manual, respectively.

The authors appreciated discussions about geophysics-geology integration with L. Ailleres and the rest of the Loop consortium's researchers. The authors thank Guillaume Pirot for his comments on the manuscript. The authors are also thankful to Mahtab Rashidifard, Nuwan Suriyaarachchi, Damien Ciolczyk and Marina Zarate-Jeronimo for interesting discussions.

## 835 10 Authors' contribution

JG designed the geophysical study and ran all inversions shown in the paper. He adjusted the geological model presented here, which was initially built by MJ and ML. JG performed posterior analysis and interpretation of results. JG is the main contributor to the writing of this article and preparation of the assets. JG carried out the scaling tests shown in Appendix with the collaboration of VO.



VO and JG worked together on the implementation of the gravity and magnetic inversion methodologies in Tomofast-x. VO is the main developer of this version of Tomofast-x, the development of which was carried in collaboration with JG and RM. The main contributors to the code are VO, JG, and RM. JG performed the initial testing of the different data integration techniques presented to the exception of the cross-gradient technique, which was implemented independently by VO.

RM participated in the redaction of this paper. RM initiated the Tomofast project and implemented the initial LSQR solver for gravity inversion, which was subsequently used as a starting point for the current version of the code. RM and VO added the possibility of performing data inversion using an Lp norm ($1 < p \leq 2$) smallness term. RM and VO developed and tested the electrical capacitance tomography component of Tomofast.

       JG and RM explored the functionalities of Tomofast-x and performed extensive testing for robustness and
validation of techniques implemented, especially the cross-gradient technique.

       ML and MJ produced the reference geological model from field measurements and carried out probabilistic geological modelling. ML and MJ were involved in the redaction of the manuscript and participated in the supervision of the project. MJ and ML have been involved in the validation of the methodology at the initial development stage and supervised the overall progress of the presented work.

JG supervised work relating to the preparation of Tomofast-x's user manual and the reduced 2D version written in Python. All authors participated in the revisions of the manuscript.

**Appendix A.**  Other functionalities of Tomofast-x

A.1    Posterior uncertainty indicators

**A.1.1    Posterior LSQR variance matrix**

At the first and last iteration of the inversion, the diagonal elements of the posterior covariance matrix of the recovered model is calculated in Tomofast-x. This calculation is performed following (Kostina et al., 2009) who introduce an extension of the LSQR algorithm where such matrix is calculated at each iteration. The variances are part of the outputs of Tomofast-x for further analysis by the user, such as the estimation of uncertainty in the recovered property model.

**A.1.2    Jacobian of the cost function**

       Tomofast-x offers the possibility to examine the Jacobian matrix of the misfit function (eq. 5), which encapsulates the contribution of several constraint terms (see for example eq. 5), by calculating its derivative with respect to the model $m$, $\frac{\partial \theta(d,m)}{\partial m}$. This feature takes advantage of the LSQR solver. In the LSQR algorithm, $\frac{\partial \theta(d,m)}{\partial m}$ is calculated at the beginning of each iteration when approximating a solution to the system of equations (Appendix
D). The value of $\frac{\partial \theta(d,m)}{\partial m}$ can then be calculated before or after application of the depth weighting operator. It is computed as the product of the transpose of the matrix representing the left-hand side of the system of equations to be solved by the LSQR solver, by the vector of the corresponding quantities to minimize (data misfit, cross-gradients, damping terms, etc., constituting the right hand side of the corresponding equation, as shown in





Appendix D). Importantly, its dimension is equal to the number of model parameters. It is therefore possible to
store it on disk to provide a measure of the sensitivity of the data and the different terms of the misfit function to
model variations at depth or in any part of the computational domain. By computing $\left\|\frac{\partial \theta(d,m)}{\partial m}\right\|$, it is possible to
study the convergence of the algorithm, small values indicating convergence. In addition, it is a metric that
measures the stability of the algorithm and which is useful to determine whether the system of equations is well
conditioned.

**A.1.3    Identification of rock units**

Membership analysis of the inverted model can be performed when statistical petrophysical constraints were
applied to inversion from the values of $\mathcal{N}_k$ reached after inversion converged. Membership values can be used to
assess inverted models by reconstructing a rock unit model from the recovered inverted physical properties
(Doetsch et al., 2010, Sun et al., 2012, Giraud et al., 2019c). Rock units labels can also be assigned to model-cells
when the ADMM bound constraints have converged. It allows attaching a petrophysical property interval to each
model-cell, allowing direct identification of rock types.

**A.1.4    Cross-product of gradients**

The cross-product of model gradients in 3D can be stored after inversion and its $L_2$ norm is given after each
inversion cycle. It allows to assess the degree of structural similarity between the models, and to delineate areas
showing specific structural similarities or dissimilarities.

A.2    Jacobian matrix truncation

Tomofast-x offers the possibility to use a moving sensitivity domain approach (Čuma et al., 2012, Čuma and
Zhdanov 2014) limiting the sensitivity domain to a cylinder which radius is chosen by the user to reduce
computational requirements (the option for a sphere is also present in the source code, but commented in the
current version). The underlying assumption is that cells beyond a given distance exert a negligible influence on
the measurement. Generally speaking, this radius is provided by the users and should be chosen carefully.

**A.3    $L_p$ norm**

Tomofast-x also offers the possibility of performing data inversion using a $L_p$ norm ($1 < p \leq 2$) to define the
smallness term, as it has been proposed for electrical capacitance tomography (ECT) in (Martin et al., 2018) in
the framework of Tomofast-x. The $L_p$ norm inverse problem is non-linear and can be solved iteratively using $L_2$
minimization. In the $L_p$ norm case, the regularization parameter can be approximated by a p-power law of the
model at each point of the computational domain and must also be recomputed at each new inversion cycle. When
the $L_p$ norm is introduced for $p < 2$, this procedure allows to obtain sharper models with better interface definition,
and determine stronger contrasts for the specific cases under study. If $p = 2$, the smallness term is reduced to $L_2$
norm minimization (the commonly-used Tykhonov-like regularization) as used in this work. The choice of other
values such that $1 < p \leq 2$ is at the discretion of the user or depending on prior information.



### A.4 Electrical capacitance tomography

Detailing electrical capacitance tomography (ECT) is beyond the scope of this paper, but we can apply to joint gravity and magnetic inversion the functionalities that have been introduced to solve the ECT inverse problem

based on $L_2$ data misfit norm and $L_p$ ($1<p\leq2$) damping term minimization (see Sect. A.3). In Tomofast-x, ECT is based on the finite-volume method for the forward problem and on a non-linear and iterative LSQR method to solve the inverse problem. As in propagative and diffusive geophysical inverse problems in frequency domain, the sensitivity matrix and the damping term depend on the current model and must be recomputed at each new iteration. We refer the reader to Martin et al. (2018) for more details on this technique. Note that algorithms

developed in relation to this method can easily be extended to propagative and diffusive geophysical inverse problems.

**Appendix B.** Summary of the notation and terms used in the paper

| Symbol | Definition |
|---|---|
| Subscripts and superscripts | |
| $d$ | Refers to 'data': 'mag' or 'grav'. |
| $m$ | model |
| $pr$ | Refers to 'prior' |
| $g$ | gradient |
| $x$ | Cross-gradient |
| $pe$ | Refers to 'petrophysics' |
| a | ADMM |
| G | Gravity |
| M | Magnetics |
| Model and physical quantities | |
| $\boldsymbol{m}$ | Property model inverted for |
| $\boldsymbol{v}$ | ADMM variable |
| $\boldsymbol{u}$ | ADMM variable |
| $\omega$ | Membership value in Gaussian mixture |
| $\psi$ | Membership value in Gaussian mixture (from geology) |
| $\boldsymbol{\mu}$ | Mean value of petrophysical properties |
| $\boldsymbol{\sigma}$ | Standard deviation of petrophysical properties |
| $\boldsymbol{z}$ | ADMM variable |
| $\varepsilon$ | Positive threshold real number such that $z \gg \varepsilon$ |
| $n_{data}$ | Number of geophysical data points |
| $n_f$ | Number of rock formations |
| $\boldsymbol{d}$ | Geophysical data |



| β | Exponent for depth weight power law |
|---|---|
| Mathematical operators or notations | |
| $g(.)$ | Geophysical forward operator |
| $P(\cdot)$ | Petrophysical distribution operator |
| $\nabla \cdot$ | Gradient operator |
| $\mathbf{L_p}$ | L-p norm |
| $\mathbf{L_2}$ | L-2 norm (sum-of-squares) |
| $\boldsymbol{S}$ | Geophysical data sensitivity matrix |
| Diagonal matrices $\boldsymbol{W}$ | |
| $\boldsymbol{W_d}$ | diagonal matrix; all elements equal to the sum-of-squares of the data |
| $\boldsymbol{W_m}$ | Smallness term covariance matrix |
| $\boldsymbol{W_g}$ | Smoothness term covariance matrix |
| $\boldsymbol{W_x}$ | Cross-gradient term covariance matrix |
| $\boldsymbol{W_{pe}}$ | Petrophysical term covariance matrix |
| $\boldsymbol{W_a}$ | ADMM term covariance matrix |
| $\boldsymbol{D}$ | Depth weighting operator |
| Weighting terms $\alpha$ | |
| $\alpha_m$ | model |
| $\alpha_g$ | Gradient |
| $\alpha_x$ | Cross-gradient |
| $\alpha_{pe}$ | petrophysics |
| $\alpha_a$ | Multiple bounds constraints |
| $\alpha^G$ | Weight assigned to the gravity inverse problem (used only in joint inversion) |
| $\alpha^M$ | Weight assigned to the magnetic inverse problem (used only in joint inversion) |

**Appendix C.** Forward gravity and magnetic data calculation

In this Appendix we summarize the forward calculation of gravity and magnetic data as performed in Tomofast-x. In practice, Tomofast-x calculates forward data using input data expressed in units from *Système International* (SI), expressed in kilogram, metre, and second. The gravity field $\boldsymbol{f}$ of a distribution of density anomalies $\Delta\rho$ over a volume of rock $V$ at a location $\boldsymbol{r}' = [x', y', z']$ can be expressed as follows:

$$\boldsymbol{f}(\boldsymbol{r}) = G \iiint_V \Delta\rho(\boldsymbol{r}) \frac{\boldsymbol{r} - \boldsymbol{r}'}{|\boldsymbol{r} - \boldsymbol{r}'|^3} dV \qquad (24)$$

Where $\boldsymbol{r} = [x, y, z]$ defines the location of mass density anonaly $\Delta\rho(\boldsymbol{r})$ and $G$ is the universal gravity constant. While Tomofast-x is implemented in such a way that the three spatial components of $\boldsymbol{f}$ can be obtained, we consider only the vertical direction here, which we simply write $f$ for sake of clarity (note that here, when taken



for the whole model, $f = \boldsymbol{g_G}$). In our implementation, the volume $V$ is descretized in $N_m$ rectangular prisms (model cells) of constant density. Discretised, equation 26 then rewrites as:

$$f(x,y,z) = G \sum_{k=1}^{N_m} \Delta\rho_k \frac{z_k - z'}{((x-x_k)^2 + (y-y_k)^2 + (z-z_k)^2)^{\frac{3}{2}}} \Delta x_k \Delta y_k \Delta z_k, \tag{25}$$


where $\Delta x_k$, $\Delta y_k$ and $\Delta z_k$ define the dimensions of the $k^{\text{th}}$ rectangular prism. In our discretization, we assume a model constituted of $n_x \times n_y \times n_z$ cells, with $n_x$, $n_y$ and $n_z$ representing the number of cells in each direction. This leads to the computation of $f$ using the following formulation of equation 26:

$$f(x,y,z) = \sum_{i=1}^{n_x}\sum_{j=1}^{n_y}\sum_{k=1}^{n_z} \Delta\rho_{i,j,k} S_{i,j,k} \tag{26}$$

Where, using the formulation of (Blakely, 1995), the elements of the sensitivity matrix $S$ are given as:

$$S_{i,j,k} = G \sum_{m=1}^{2}\sum_{q=1}^{2}\sum_{t=1}^{2} (-1)^{m+q+t} \left[ \zeta_t \operatorname{atan}\left(\frac{\xi_m \eta_q}{\zeta_t R_{m,q,t}}\right) - \xi_m \log(R_{m,q,t} + \eta_q) - \eta_q \log(R_{m,q,t} + \xi_m) \right], \tag{27}$$

where, $\xi_m$, $\eta_q$ and $\zeta_t$ are the coordinates of the vertices of the prism, and

$$R_{m,q,t} = \left[ (x-\xi_m)^2 + \left(y-\eta_q\right)^2 + (z-\zeta_t)^2 \right]^{\frac{1}{2}} \tag{28}$$

In Tomofast-x, the total magnetic field anomaly is calculated by summing the responses of the prisms making up the model, following Bhattacharyya (1964, 1980). The Regional magnetic field is denoted $F = \left(F_x, F_y, F_z\right)$, and

the Magnetization $\boldsymbol{M} = \left(M_x, M_y, M_z\right)$. We write $F = \|\boldsymbol{F}\|$ and $M = \|\boldsymbol{M}\|$. Note that remnant magnetization is not accounted for.

Using the formalism of (Blakely, 1995), we denote $\Delta T$ the magnitude of the total magnetic field anomaly generated by a prism oriented parallel to the $x$, $y$, and $z$ axes of the mesh similarly to the gravity case. We have:

$$\Delta T(x,y,z) = \sum_{i=1}^{n_x}\sum_{j=1}^{n_y}\sum_{k=1}^{n_z} \chi_{i,j,k} S_{i,j,k} \tag{29}$$

where $\chi$ is the magnetic susceptibility. The sensitivity $S$ is given as:


$$S_{i,j,k} = \mu_0 F \sum_{m=1}^{2} \sum_{q=1}^{2} \sum_{t=1}^{2} (-1)^{t+1} \left[ \frac{\alpha_{yz}}{2} \log\left( \frac{R_{m,q,t} - \xi_m}{R_{m,q,t} + \xi_m} \right) + \frac{\alpha_{xz}}{2} \log\left( \frac{R_{m,q,t} - \eta_q}{R_{m,q,t} + \eta_{q_m}} \right) \right.$$
$$- \alpha_{xy} \log(R_{m,q,t} + \zeta_t) - M_x F_x \, \text{atan}\left( \frac{\xi_m \eta_q}{\xi_m^2 + R_{m,q,t}\zeta_t + \zeta_t^2} \right) \qquad (30)$$
$$\left. - M_y F_y \, \text{atan}\left( \frac{\xi_m \eta_q}{R_{m,q,t}^2 + R_{m,q,t}\zeta_t - \xi_m^2} \right) + M_z F_z \, \text{atan}\left( \frac{\xi_m \eta_q}{R_{m,q,t}\zeta_t} \right) \right],$$


where, $\xi_{m=1,2}$, $\eta_{q=1,2}$ and $\zeta_{t=1,2}$ are the coordinates of the vertices of the prism along the $x$, $y$, and $z$ directions, respectively. The other terms of equation 30 are defined below:

$$\alpha_{xz} = F_x M_z + F_z M_z, \qquad (31)$$

$$\alpha_{xy} = F_x M_y + F_y M_x, \qquad (32)$$

$$\alpha_{yz} = F_y M_z + F_z M_y. \qquad (33)$$

Scaling tests

Although Tomofast-x can run on personal computers in a few seconds or minutes for 2D inversions and small 3D
volumes (typically a few minutes on a laptop for models smaller than approx. 100,000 model cells), it necessitates
a supercomputer for realistic size 3D case studies (e.g., models exceeding 500,000 model cells and 10,000
geophysical data points).

We assess Tomofast-x's parallel efficiency using the strong scaling as an indicator. The strong scaling curve is
given by plotting the number of CPUs as a function of user time. It is complemented by the relative speedup curve
$t_1/t_{n_{cpu}}$, where $t_1$ and $t_{n_{cpu}}$ are respectively the user times to complete inversion using the number of CPUs
$n_{cpu} = 1$ and a given number of CPU $n_{cpu}$, respectively. We performed the scaling tests on the EOS machine
from the CALMIP supercomputing centre (https://www.top500.org/site/50539/, https://www.calmip.univ-
toulouse.fr/spip.php?article388 – the latter being in French only, both last accessed on 10/11/2020).

The full-size model we used is made of $N_x N_y N_z = 128 * 128 * 32 = 524{,}288$ cells (i.e., $2^{19}$ cells), which we
reduce by a factor of 2 by reducing the physical domain incrementally to $N_x N_y N_z = 32 * 32 * 32 = 32{,}768$ cells
(i.e., $2^{15}$ cells) to be able to use it on a single CPU, for the purpose of parallelization efficiency analysis. In the
configuration we use, the number of data points modelled is equal to: $N_{data} = N_x N_y$.

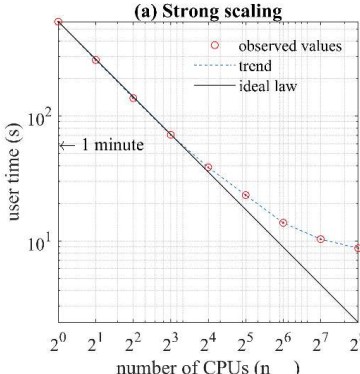
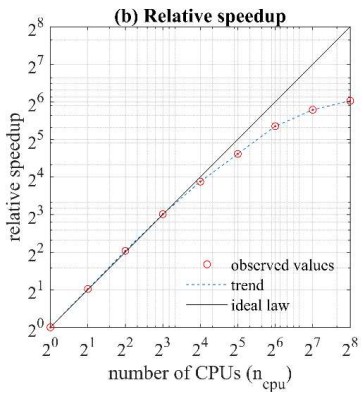

**Figure A 1 - Strong scaling (a), relative speedup (b) and number of elements per CPU (c) plots for a number of cpus**
**equal to 1, 2, 4, 8, 16, 32, 64, 128, 256, and 512. in (c), the intersection. between the two curves symbolises the number of elements below which computational resources usage is suboptimal. The line marked 'ideal law' indicates perfect scalability for the tests that were performed.**

Figure A 1a shows the parallelization efficiency. It reveals that the scaling is nearly perfect for up to 16 CPUs, very good for 32 CPUs and that it deteriorates above 64 CPUs. This corresponds to relative speedups (ratio
between run time for one CPU and a given number of CPUs) of about 14.5, 25 and 40(Figure A 1b), respectively. For the cases using 64, 128 and 256 CPUs, speedup increases from 40 to 65, indicating that overhead inter-processor communication time for $n_{cpu} \geq 64$ increasingly impacts the total computation time, for this (small) problem size. This is noticeable in Figure A 1a and Figure A 1b as both curves seem to adopt an asymptotic behaviour for the largest numbers of CPUs. This illustrates the deterioration of performances due to inter-
processor communications (Kumar et al., 1994). The deterioration of performances due to inter-processor communications is due to the number of elements (or model cells) processed by each CPU becoming smaller, while the number of elements involved in communications increases; ultimately, the time spent in pure computations in each core becomes smaller than the time spent in inter-processor communications.

The efficiency curves (e.g., Figure A 1a and Figure A 1b) allow us to determine the minimum number of elements
per CPU that run efficiently (Hammond and Lichtner 2010). The case of $n_{cpu} = 64$ marks an inflection point in Figure A 1b, corresponding to point of diminishing return equal to a number of element per CPUs of 512. This indicates that for this particular configuration, it is preferable to run inversions with $n_{cpu} \leq 64$ to maximize parallel efficiency. For better understanding and interpretation of scaling and speedup, we remind the number of elements $n_{el}$ per CPU as a function of $n_{cpu}$:

$$n_{el} = N_x N_y N_z / n_{cpu}. \tag{34}$$

As a general rule, we recommend to respect the condition of $n_{el} \geq 512$. For smaller number of elements, the allocated resources are used in a suboptimum manner. Note that the memory requirements vary proportionally with $N_x N_y N_z N_{data} = n_m n_d$.

### Appendix D. Matrix formulation of least-squares problem and resolution of the inverse problem

This appendix introduces the matrix formulation of equation 5 and its resolution.



We can write the system of equations to be solved in the least-squares sense as:

$$
\begin{bmatrix}
\alpha_d \boldsymbol{Sm} \\
\alpha_m \boldsymbol{W_m m} \\
\alpha_g \boldsymbol{W_g} \nabla \boldsymbol{m} \\
\alpha_{pe} \boldsymbol{W_{pe} P(m)} \\
\alpha_x \boldsymbol{W_x} (\nabla \boldsymbol{m_1} \times \nabla \boldsymbol{m_2}) \\
\alpha_a \boldsymbol{W_a m}
\end{bmatrix}
=
\begin{bmatrix}
\alpha_d \boldsymbol{d} \\
\alpha_m \boldsymbol{W_m m_{pr}} \\
0 \\
\alpha_{pe} \boldsymbol{W_{pe} P_{max}} \\
0 \\
\alpha_a \boldsymbol{W_a (v - u)}
\end{bmatrix}
\tag{35}
$$

At iteration $k$, the system of equation is linearized around the current model. It is solved for the optimum update of the current model $m^k$ model update as described below. Models $\boldsymbol{m_1}$ and $\boldsymbol{m_2}$ are set accordingly with the type of inversion considered.

In Tomofast-x, depth weighting $D$ is applied as a sensitivity matrix preconditioner. The resulting system is solved using the LSQR algorithm (Paige and Saunders 1982) as:

$$
\begin{bmatrix}
\alpha_d \boldsymbol{SD^{-1}} \\
\alpha_m \boldsymbol{W_m} \\
\alpha_g \boldsymbol{D^{-1} W_g} \dfrac{\partial \nabla \boldsymbol{m_k}}{\partial \boldsymbol{m}} \\
\alpha_{pe} \boldsymbol{D^{-1} W_{pe} P'(m_k)} \\
\alpha_x \boldsymbol{D^{-1} W_x} \dfrac{\partial}{\partial \boldsymbol{m}} (\nabla \boldsymbol{m_1} \times \nabla \boldsymbol{m_2}) \\
\alpha_a \boldsymbol{D^{-1} W_a}
\end{bmatrix}
\Delta \overline{m}_{k+1} = -
\begin{bmatrix}
\alpha_d (\boldsymbol{d} - g(\boldsymbol{m^k})) \\
\alpha_m \boldsymbol{W_m (m_k - m_{pr})} \\
\alpha_m \boldsymbol{W_g} \nabla \boldsymbol{m_k} \\
\alpha_{pe} \boldsymbol{W_{pe} (P(m_k) - P_{max})} \\
\alpha_x \boldsymbol{W_x (\nabla m_k \times \nabla m_x)} \\
\alpha_a \boldsymbol{W_a (m_k - z^k + u^k)}
\end{bmatrix}
\tag{36}
$$

At each $k^{\text{th}}$ inversion cycle, we solve this system of equations and calculate the model update the model as follows:

$$
\Delta \boldsymbol{m_{k+1}} = \boldsymbol{D^{-1}} \Delta \overline{m}_{k+1}.
\tag{37}
$$

The model $\boldsymbol{m^k}$ can then be updated to obtain $\boldsymbol{m^{k+1}}$.

$$
\boldsymbol{m_{k+1}} = \boldsymbol{m_k} + \Delta \boldsymbol{m_{k+1}},
\tag{38}
$$


Following (Ogarko et al., 2020), $u^0 = 0$, $z^0 = 0$. The updated ADMM variables $\boldsymbol{z^{k+1}}$ and $\boldsymbol{u^{k+1}}$ are calculated using the ADMM algorithm introduced by Boyd (2010):

| $\boldsymbol{z^{k+1}} = \pi_B (\boldsymbol{m_{k+1}} + \boldsymbol{u^k})$, | (39) |
|---|---|
| $\boldsymbol{u^{k+1}} = \boldsymbol{u^k} + \boldsymbol{m_{k+1}} - \boldsymbol{z^{k+1}}$, | (40) |

where $\pi_B$ is a projection onto the bounds $B$ such that:

| $\pi_B(x) = [\pi_{B_1}(x_1), \pi_{B_2}(x_2), \dots, \pi_{B_n}(x_n)]$, with | (41) |
|---|---|
| $\pi_{B_i}(x_i) = \arg \min\limits_{y \in B_i} \|x_i - y\|_2$ | (42) |

The value of the starting model $\boldsymbol{m^0}$ is provided by the user.





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
