# Peer review of "Structural, petrophysical and geological constraints in potential field inversion using the Tomofast-x v1.0 open-source code"

_Geoscientific Model Development, 2021_

## Referee Comment (RC2)

[referee-annotated manuscript omitted]

---

## Author Response (AR1)

Dear Revision Committee,

Thank you for contribution to the revision of our manuscript.

We have address the reviewers' comments and posted our answers. Below is a consolidated view of our answer.

Best regards,

The authors

ANSWER TO RC1

Dear Reviewer,

Thank you for taking the time to review our submission. We are pleased to read that you are of the opinion that it has reached its objective. We have started correcting the minor language mistakes and typos you mention and will proceed carefully with the next stage of the review and publication process.

Thanks, and regards,

The Authors

ANSWER TO RC2

Dear Mehrad Bastani,

Thank you for your review and for your comments. We are pleased to read that you enjoyed reading our work!
Below are our answer to the your comments.

Reviewer:
"I would like to see how meaningful the variation of density contrast at a depth of 10 km or deeper can be when modelling the gravity data and it is the same with the magnetic field data? A presentation of sensitivty analysis can be a good way to test/answer this point."

Answer:
One of the options offered by Tomofast-x is to write the diagonal values of the sensitivity matrix multiplied by its transpose to the hard drive after completion of the inversion as part of the outputs. From there, one can then estimate the sensitivity of the data to variations in any model cell. We agree that the kind of sensitivity study you mention is interesting but we prefer not to propose it in the manuscript to maintain focus as the paper is already a bit long. The sensitivity of magnetic and gravity data being different due to the physics of the phenomena each method relies on, their sensitivity decays differently with the distance to the source.

We have followed most suggestions you made in the edited version of the manuscript. Our point-by-point answers to the edits you made are in the supplementary material.

Thanks and regards,

The authors

SUPPLEMENTARY MATERIAL

Answer to comments (RC2)

16/07/2021

Line 364:

**Reviewer**: "The values are not distinctly high!! Are these based on the anomalies and measured susceptibility/densities on the outcrops? It seems they are rather smooth and follow the anomaly maps."

**Answer**: Yes, these values were taken from field outcrops when available. They were otherwise selected to pose a greater challenge to the inversion. See lines 356-360 (original in black, added text in blue): "In addition to the modification of the structural model, we make adjustments on the original density values derived from field petrophysical measurements by reducing the differences between the density contrasts of different rock units. Doing so, we increase the interpretation ambiguity of inversion results and decrease the differentiability of the different rock units. The same procedure is applied to magnetic susceptibility to make accurate imaging using inversion  more challenging."

Figure 3. Thank you for pointing this out. The values in Figure 2 are indeed low: these values correspond to another model we tested. It was a mistake to show them here. We have corrected this error and changed the Figure accordingly.

Line 503:

**Reviewer**: "Do you suggest any specific approach? how important is the role of these parameters?"

**Answer**: In this work, we have opted for an empirical approach using the L-curve principle. There exist methods to automate the process (see, for instance, Farquharson and Oldenburg (2004), who compare two methods capable of automatically determining the regularisation parameter). We have not used any of them simply because their implementation lies beyond the scope of this work.

The range of values we tested were selected on purpose to be too large for the upper limit and too small for the lower limit. In other cases, we might recommend to sample the parameters in the range of values that can be estimated by simple calculations.

Given the number of regularisation parameters that can be involved when using several regularisation terms, the approach we follow here can be time consuming or computationally expensive for large datasets. We recommend using an automatic approach in such cases.

The regularisation parameters are important as they might also be used as a way to reflect the confidence one has in the hypotheses underlying the utilisation of a specific regularisation term. For instance, the value of the weight applied to the cross-gradient regularisation term may reflect the geologists' confidence in the necessity to enforce structural similarity between models. The same reasoning may also be applied to scenario testing.

In the specific examples we show, in most cases, a small change in their values (e.g,. multiplication by 1.5) may not alter the inversion results dramatically. Such values still need to be selected with care as altering the regularisation parameter by an order of magnitude or more will affect inversion results significantly.

Figure 17:
Reviewer: "A figure with data fit for each model is good to present"
Answer: all inversions have the same data misfit (or very nearly so). The datafit for all inversions is shown below in Figure A 1.

[Figure]

*Figure A 1. Datafit for all inversions shown in Figure 17.*

We mention this in the updated version the manuscript but we prefer not to add the Figure in the manuscript to avoid lengthening it. We have added the following in line 698: "We remind that all models shown here produce a similar data misfit $\Theta_d^{obj}$ accordingly with equation 19".

Line 713: Reviewer: "Do you mean this the expert input to speculate after having gained some expeience?"

Answer: Yes, we mean that while one can place a certain degree of confidence in the ranking we propose, it remains somehow speculative. We have tested the different inversions on a specific structural model, with selected petrophysical values, using geological prior information reflecting average uncertainty in the field geological orientation data. We have not sampled the different case scenarios that can be encountered in other studies, such as oil and gas exploration, hydrology, deep crustal studies, etc. For this reason, we treat our interpretation of results in terms of ranking of the regularisation parameters with care to avoid an over-generalisation of what we infer.

**References:**

Farquharson, C. G., and D. W. Oldenburg, 2004, A comparison of automatic techniques for estimating the regularization parameter in non-linear inverse problems: Geophysical Journal International, **156**, 411–425.

---

## Author Response (AR2)

Subject: Erratum on previous answer to comments

Dear Reviewer,

In this erratum, we would like to point out additional modifications we brought to the manuscript in relation to questions you posted in your review.

In answer to your comment about line 501 in the original manuscript: "Do you suggest any specific approach? how important is the role of these parameters?", we have added the following text in line 203 of the modified manuscript:

"The different $\alpha$ terms are trade-off parameters that control the importance given to the different terms during the inversion. These terms therefore play an important role and need to be determined carefully (see Sect. 4.1 and 4.2 for more details)."

We have also added the following in Section 4.1, line 481 of the revised manuscript:
"We chose this approach for its simplicity and note that there exist other techniques that use an automated process, such as the generalized cross-validation technique (Craven and Wahba, 1978)."

In answer to your comment about line 713 in the original manuscript: "Do you mean this the expert input to speculate after having gained some expeience?", we have added the following text in line 718 of the modified manuscript:

"We note that this ranking remains speculative as it might apply only to models sharing similarities with the case we investigated."

References:

Craven, P. and Wahba, G.: Smoothing noisy data with spline functions, Numer. Math., 31, 377–403, doi:10.1007/BF01404567, 1978.